

# Radiative-Convective Equilibrium Model Intercomparison Project

Allison A. Wing [1], Kevin A. Reed [2], Masaki Satoh [3], Bjorn Stevens [5], Sandrine Bony [4], and Tomoki Ohno [3]

[1]Florida State University, Tallahassee, FL USA
[2]Stony Brook University, Stony Brook, NY USA
[3]Atmosphere and Ocean Research Institute, University of Tokyo, Kashiwa, Japan
[4]Laboratoire de Météorologie Dynamique, IPSL, CNRS, Paris, France
[5]Max Planck Institute for Meteorology, Hamburg, Germany

*Correspondence to:* Allison A. Wing (awing@fsu.edu)

**Abstract.** *RCEMIP*, an intercomparison of multiple types of models configured in radiative-convective equilibrium (RCE), is proposed. RCE is an idealization of the climate system in which there is a balance between radiative cooling of the atmosphere and heating by convection. The scientific objectives of *RCEMIP* are three-fold. First, clouds and climate sensitivity will be investigated in the RCE setting. This includes determining how cloud fraction changes with warming and the role of self-

5 aggregation of convection. Second, *RCEMIP* will quantify the dependence of the degree of convective aggregation and tropical circulation regimes on temperature. Finally, by providing a common baseline, *RCEMIP* will allow the robustness of the RCE state, cloud feedbacks, and convective aggregation across the spectrum of models to be assessed. A novel aspect and major advantage of *RCEMIP* is the accessibility of the RCE framework to a variety of models, including cloud-resolving models, general circulation models, global cloud-resolving models, and single column models.

**1 Introduction**

Radiative-convective equilibrium (RCE) has long been used as an idealization of the climate system. In a greenhouse atmosphere convection must balance the radiative heat-loss of the atmosphere making radiative-convective equilibrium the simplest possible description of the climate system (Dines, 1917). For this reason there is a rich history of modeling RCE, mostly as a one dimensional problem (e.g., Möller, 1963; Manabe and Strickler, 1964; Satoh and Hayashi, 1992; Renno et al., 1994).

These early studies of RCE helped formulate an understanding of the basic characteristics of climate and the first estimates of climate sensitivity (Ramanathan and Coakley, 1978; Charney et al., 1979). Early work with two-dimensional simulations of RCE studied the relationship between convection and environmental structures (Nakajima and Matsuno, 1988; Held et al., 1993; Sui et al., 1994; Randall et al., 1994; Grabowski et al., 1996). In recent years, as it became possible to perform more computationally intensive numerical simulations of RCE, there has been a revival in the use of RCE to study a variety of prob-

lems in tropical meteorology and climate. One common configuration is to simulate RCE with a three-dimensional numerical model with explicitly resolved convection over domain lengths of 100-1000 km (e.g., Tompkins and Craig, 1998; Bretherton et al., 2005). The RCE state is achieved by prescribing uniform solar insolation and a horizontally uniform boundary condition (constant sea surface temperature (SST) or a slab ocean model) and initializing the model with random noise. There is also a





growing body of work employing RCE in general circulation models (GCMs) with parameterized clouds and convection (e.g., Held et al., 2007; Popke et al., 2013; Bony et al., 2016).

The popularity of the RCE arises from the fact that it remains the simplest way to phrase many important questions about the climate system. RCE has been extensively used to help answer questions such as how the representation of subgrid-scale

processes influence the coupling of clouds and convection to the climate system (e.g., Satoh and Matsuda, 2009; Becker et al., 2017), and how this coupling depends on temperature (e.g., Muller et al., 2011; Romps, 2011; Singh and O'Gorman, 2013, 2014, 2015; Seeley and Romps, 2015, 2016; Hohenegger and Stevens, 2016). RCE has been used to study the predictability of mesoscale rainfall (e.g., Islam et al., 1993), tropical anvil clouds (Bony et al., 2016; Seeley et al., 2017; Cronin and Wing, 2017), precipitation extremes (e.g., Muller, 2013; Singh and O'Gorman, 2014; Pendergrass et al., 2016), as well as how the

land surface influences the climate state (Becker and Stevens, 2014), or the rectifying effects of surface heterogeneity in the form of islands (e.g., Cronin et al., 2015). RCE has been used as a background state for tropical cyclone studies (e.g., Nolan et al., 2007; Chavas and Emanuel, 2014; Reed and Chavas, 2015). A central theme arising in many of these studies, and related to the formation of tropical cyclones (Wing et al., 2016) and the Madden-Julian Oscillation (Arnold and Randall, 2015; Satoh et al., 2016), is the role of convective aggregation, which often arises spontaneously in studies of RCE using explicit and

parameterized convection (Wing et al. (2017), and references therein). It remains an open question as to how and whether the real atmosphere self aggregates, and to what extent this is important for the properties of the climate system (Bony et al., 2015) in part because these aspects of the simulations appear sensitive to how the models are formulated (e.g., Muller and Held, 2012).

Assessing the formulaic sensitivity of simulations of RCE is hindered by the absence of a common baseline. Past studies

differ in many, seemingly unessential details, which makes them hard to compare. These range from different prescriptions of boundary conditions, such as incoming solar radiation, to different specifications of atmospheric composition, to different treatments of the upper atmosphere, or surface properties such as albedo. To provide context for the many studies that have been performed so far, and to provide a starting point for the many studies to come, a common baseline would be helpful. In this paper we propose such a baseline in the form of a model intercomparison study, *RCEMIP*. A standard configuration of

RCE is a useful framework for model development and evaluation (Held et al., 2007; Reed and Medeiros, 2016), but in addition to providing a fixed point for past and future studies, such an intercomparison can itself address important questions related to RCE. It may also help extend the range of models used to simulate RCE, for instance by including single-column model setups. Already groups are beginning to compare RCE solutions using general circulation models with parameterized physics on large-domains, to simulations on smaller domains with finer grids, to solutions using cloud resolving models (Silvers et al.,

2016; Hohenegger and Stevens, 2016). No other framework is accessible by so many of the varied models used to study the climate system, as in addition to cloud-resolving models, general circulation models, and single column models, even Earth System Models of Intermediate Complexity (Claussen et al., 2002) could be applied to the problem of RCE. Hence through the definition of a common baseline it should be possible to encourage the study of this canonical representation of the climate system using an even wider range of models. In addition to the simplicity and accessibility of the RCE framework,

its importances lies in its similarities to substantial aspects of the real atmosphere; in general, RCE states are thought to



correspond to convective regions over the tropical western Pacific warm pool, in terms of thermal structure. There have already been some efforts to consider RCE simulations within a hierarchy of models; for example, Popke et al. (2013) found similar cloud feedbacks between a GCM in a realistic configuration and in RCE and Satoh et al. (2016) compared the structure of tropical convective systems in between Earth-like aquaplanet experiments and RCE. A standard configuration of RCE would

enable more of these types of comparisons.

Given this backdrop in what follows we describe the proposed model intercomparision study, *RCEMIP* and more specifically detail the questions it will be used to address. In Sect. 2 we state the main scientific questions that this intercomparison will address. Subsequent sections specify the experimental design, including the required output and diagnostics. Finally, to give a flavor and better guide those who wish to participate in this study, in Sect. 5 we present some sample results from a cloud-

resolving model and a GCM.

## 2   Science Objectives

The three themes that *RCEMIP* has been designed for are:

1. **What is the response of clouds to warming and what is the climate sensitivity of RCE?**

2. **What is the dependence of the degree of convective self-aggregation and tropical circulation regimes on tempera-**

**ture?**

3. **What is the robustness of the RCE state, cloud feedbacks, and convective self-aggregation across the spectrum of models?**

The first theme of *RCEMIP*, clouds and climate sensitivity, is motivated by the fact that cloud feedbacks are the largest source of uncertainty in estimates of climate sensitivity and depend on processes that are largely parameterized in global climate

models (e.g., Boucher et al., 2013). The role of convection in cloud feedbacks is central to a better understanding of global and regional climate changes, as pointed out by the WCRP Grand Challenge on Clouds, Circulation, and Climate Sensitivity (Bony et al., 2015). *RCEMIP*, which includes both CRMs and GCMs, is uniquely situated to determine the response of certain types of clouds to warming, without the complications of topography, latitudinal insolation gradients, and the associated dynamical disturbances. For instance, recent work has suggested a thermodynamic mechanism for a decrease in anvil cloud fraction with

warming in several GCMs (Bony et al., 2016) and a CRM (Cronin and Wing, 2017), but the robustness of this response across a wider range of models has yet to be determined. For example, one other CRM finds the opposite response, an increase in anvil cloud fraction with warming (Singh and O'Gorman, 2015). Changes in the amount and height of anvil clouds with warming have strong implications for cloud feedbacks, and the coupling between temperature, cloud amount, and circulation may contribute to a narrowing of convective areas – both of which could lead to a type of iris effect (Mauritsen and Stevens,

2015; Bony et al., 2016; Byrne and Schneider, 2016, 2017). Given the changes in cloud fraction with warming, the climate sensitivity (or net feedback parameter) of the RCE state can be computed. This is reminiscent of the use of single column model simulations of RCE for the very first estimates of climate sensitivity, but now RCE can be simulated in much more





advanced models that allow relative humidity and clouds to vary, including models that allow for the generation of large-scale circulations by self-aggregation. The climate sensitivity of RCE simulations would reflect the fundamental characteristics of each model's representation of tropical clouds and convection, as opposed to CMIP-type simulations, which include many additional complexities such as ice-albedo feedbacks and interactions between clouds and midlatitude baroclinic eddies. A

potentially important factor in determining the climate sensitivity of RCE is the extent to which a given model's convection self-aggregates and how the aggregation changes with warming. Self-aggregation has been hypothesized to be important for climate and climate sensitivity (Khairoutdinov and Emanuel, 2010; Mauritsen and Stevens, 2015) because both numerical simulations (e.g., Bretherton et al., 2005) and observational analyses (e.g., Tobin et al., 2013) indicate the mean atmospheric state is drier and more efficient at radiating heat to space when convection is more aggregated. Much remains to be understood,

however, about how the self-aggregation in idealized simulations is borne out in the real atmosphere (Holloway et al., 2017). The role of self-aggregation in climate is therefore an aspect of climate sensitivity that *RCEMIP* will target.

The manner and extent to which self-aggregation is temperature dependent is strongly related to the impact of aggregation on climate sensitivity but remains unresolved (Wing and Emanuel, 2014; Emanuel et al., 2014; Wing and Cronin, 2016; Coppin and Bony, 2015; Bony et al., 2016). Therefore, the second theme of *RCEMIP* is about the dependence of the degree of convec-

tive self-aggregation on temperature; for instance, whether convection becomes more or less aggregated in a warmer climate. Not only does the degree of self-aggregation have implications for climate feedbacks, changing convective organization has also been shown to be one mechanism for increases in extreme precipitation with warming (Pendergrass et al., 2016). Changes in the amount of organized convection have also been linked to observed regional tropical precipitation increases (Tan et al., 2015). In addition, the fact that self-aggregation generates large-scale overturning circulations allows us to ask the more gen-

eral question of how tropical circulation regimes change with climate. In CRMs with domain geometries capable of containing multiple self-aggregated regions, there is the additional possibility of examining interactions between clouds, convection, and circulation in a framework that explicitly simulates both convection and the large-scale circulation in which it is embedded, which is a rare combination. Across both CRMs and GCMs, *RCEMIP* will be able to assess how circulation strength depends on temperature.

The final theme of *RCEMIP* is the robustness of the RCE state, its changes with warming, and representation of self-aggregation across the spectrum of models. In addition to baseline characteristics, such as the climatological anvil cloud fraction, it would be valuable to determine the universality of theoretical invariances or relationships found in a single model. For example, relative humidity has been argued to be a function of temperature only by Romps (2014) and radiative flux divergence has been found to be a nearly universal function of temperature (Ingram, 2010; Jeevanjee and Romps, 2016; Cronin

and Wing, 2017). Such invariances could simplify thinking about the response of RCE, and perhaps the actual tropics, to warming, but there is a lack of understanding of their robustness across models. In addition, a comparison of the inter-model spread in climate sensitivity of the RCE simulations with the inter-model spread of CMIP6 simulations would be informative. Despite the simplicity of the RCE setup, there is the potential for a wide range of behavior given how essential clouds and convective processes are to determining the RCE state, and the myriad of different ways these processes are represented in

models. Further, while multiple common features and mechanisms have emerged across different studies of self-aggregation





(Wing et al., 2017), the behavior of self-aggregation of convection across a wide range of models set up in a consistent manner has not been fully characterized. *RCEMIP* will enable us to truly determine the robustness of self-aggregation and its sensitivities, an important step to understanding its role in climate.

## 3 Simulation Design

The experimental design of *RCEMIP* is to require a small set of experiments that are designed to maximize the utility of the *RCEMIP* simulations in answering the questions about clouds, climate sensitivity, and self-aggregation posed above while minimizing the effort required by the modeling groups.

### 3.1 Required Simulations

We propose the following three simulations to be required for participation in *RCEMIP*:

1. `RCE295`: RCE simulation with uniform, fixed sea surface temperature (SST) of 295 K.

2. `RCE300`: RCE simulation with uniform, fixed SST of 300 K.

3. `RCE305`: RCE simulation with uniform, fixed SST of 305 K.

### 3.2 RCE Setup

RCE is simulated in a modeling setting by imposing a homogeneous lower boundary mimicking the thermodynamic state
of a sea surface with a fixed (i.e., spatially uniform) temperature and spatially uniform insolation as a forcing. The model is initialized with the same temperature and moisture sounding at every grid point and zero wind, and convection is generated by prescribing some symmetry-breaking random noise. The model is then run to stationarity, at which time irradiances, precipitation, and other variables have stopped trending up or down and exhibit variability about an approximately constant value. Here, we consider RCE in a non-rotating setting; i.e., the Coriolis parameter, $f$, or Earth's angular velocity, $\Omega$, is zero.
The mass of the atmosphere should be adjusted such that the mean sea-level pressure is 1014.8 hPa. Recommendations for geophysical constants and parameters are given in Table 1, following the convention of the Aqua-Planet Experiment (APE; http://climate.ncas.ac.uk/ape/design.html).

#### 3.2.1 Surface Boundary Condition

The lower boundary conditions is to be a spatially uniform, fixed sea surface temperature. If a skin temperature equation is
employed, the skin temperature should be equal to the prescribed surface temperature. There is no sea ice.

The surface enthalpy fluxes are to be calculated interactively from the resolved surface wind speed and air-sea enthalpy disequilibrium, enforcing a minimum wind speed of $1\,\mathrm{ms}^{-1}$ (either as $V = \max\left(V_{\mathrm{resolved}}, 1\right)$ or in quadrature as $V = \sqrt{V_{\mathrm{resolved}}^2 + 1}$). Models should compute surface exchange coefficients following their normal formulation, for instance implying stability corrections, gustiness parameterizations, or sea-state dependent roughness formations as is standard for their model tropics.





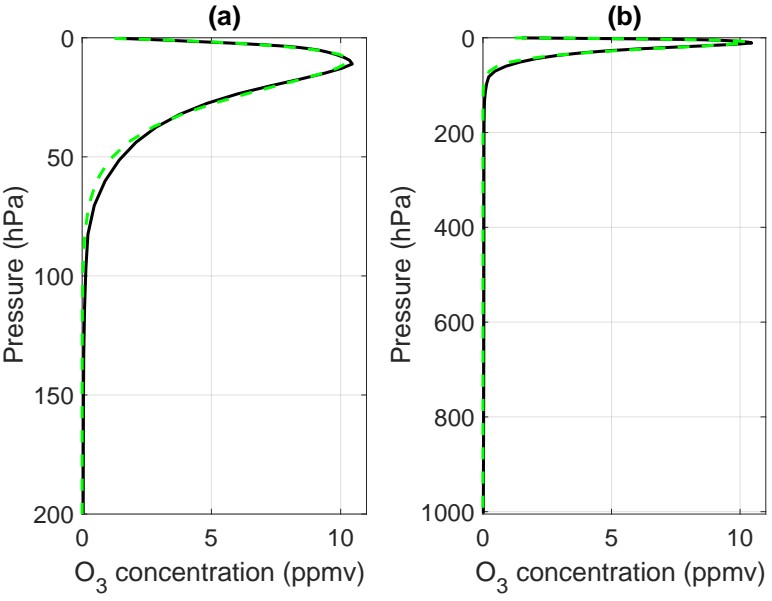

**Figure 1.** Ozone concentration (ppm) from the Aqua-Planet Experiment (black) and gamma distribution given by Eqn 1 (green-dashed), as a function of pressure above 200 hPa (a), as a function of pressure over the whole atmosphere (b).

### 3.2.2 Radiative Processes

The shortwave and longwave radiative heating rates are to be calculated interactively from the modeled state using a radiation scheme. GCMs should use the same radiation scheme as in CMIP6.

The climatologies of trace-gases are to be adjusted so that they do not have any longitudinal and latitudinal dependencies, and
their values should be fixed according to Table 2. The $CO_2$ concentration is to be set to 348 ppmv, $CH_4$ is to be 1650 ppbv, and $N_2O$ is to be 306 ppbv, following the convention of the Aqua-Planet Experiment (APE; http://climate.ncas.ac.uk/ape/design.html). Chlorluorocarbon concentrations are to be set to zero (following Popke et al. (2013)). The ozone climatology is to be an analytic approximation of the horizontally uniform equatorial profile derived from the Aqua-Planet Experiment ozone climatology (Eqn 1, Fig. 1). The ozone volumetric mixing ratio, in units of parts per million, is to be computed from pressure using a gamma
distribution:

$$O_3 = g_1 * p^{g_2} \exp\left(-p/g_3\right) \tag{1}$$

where $g_1 = 3.6478$, $g_2 = 0.83209$, $g_3 = 11.3515$, $p$ is in hPa, and $O_3$ is in ppmv.

Aerosol effects are to be ignored by zero-ing the aerosol concentrations. In some GCMs, aerosol effects may be ignored by excluding aerosol from the radiative transfer calculation and fixing the cloud droplet number concentration (we suggest
$N_c = 1.0 \times 10^8$ m$^{-3}$) and ice crystal number concentration (we suggest $N_i = 1.0 \times 10^5$ m$^{-3}$) within the microphysics parameterizations, following Reed et al. (2015). In CRMs, the cloud optical properties should be determined by the microphysics





parameterization. If specification of number concentrations or condensation nuclei is required (as in two-moment schemes), GCMs should use the Aqua-Planet configuration of their microphysics. For those models that do not have an aquaplanet config-uration (i.e., CRMs), if the microphysics scheme uses fixed cloud droplet and ice crystal number concentration, we recommend these be set to the above values $N_c$ and $N_i$. For those schemes that instead specify cloud condensation nuclei (CCN) and ice

nuclei (IN), or CCN and IN sources, they should set values consistent with the above specifications of $N_c$ and $N_i$.

Importantly, the incoming solar radiation is to be adjusted such that every model grid point sees the same incident radiation. It is to be spatially uniform and constant in time; there is to be *no* diurnal cycle and *no* seasonal cycle. A reduced solar constant of 551.58 W/m$^2$ and a fixed zenith angle of 42.05° should be used (Table 2); these values yield an insolation of 409.6 W/m$^2$, equal to the tropical (0° - 20°) annual mean. The zenith angle is equal to the average insolation-weighted zenith angle between

the equator and 20°(c.f. Cronin, 2014). The surface albedo is to be fixed at a value of 0.07, corresponding to its insolation weighted globally averaged value. As an aside, we note that if simulations with interactive surface temperature are done in the future, an implied ocean heat transport ("Q-flux") will need to be applied to prevent a runaway greenhouse effect with this value of insolation. A formulation that adjusts for heat export through the ocean is preferred to one that reduces the solar constant to mimic the combined heat transport of the ocean and atmosphere because this is believed to better represent the

competition between longwave cooling and water vapor absorption in the lower troposphere.

### 3.2.3   Initialization Procedure

*RCEMIP* will use a two-step initialization procedure. First, an RCE simulation (using the settings described above) is to be performed on a small planar domain or single GCM column. The small domain RCE simulations should be run for 100 days. This simulation should be initialized with the below analytic moisture (Eqn. (2)), temperature (Eqn. (4)) and pressure (Eqn.

(5)) profile (which approximates the observed moist tropical sounding of Dunion (2011)), and zero wind. The parameter values for the analytic profile are found in Table 2. The analytic initial specific humidity profile $q(z)$ is given, as a function of height ($z$) as

$$q(z) = q_0 \exp\left(-\frac{z}{z_{q1}}\right) \exp\left[-\left(\frac{z}{z_{q2}}\right)^2\right] \text{ for } 0 \leq z \leq z_t \qquad q(z) = q_t \text{ for } z_t < z \qquad (2)$$

where $z_t$ = 15 km approximates the tropopause height as seen in the Dunion (2011) sounding; $q_0$ is the specific humidity

at the surface ($z = 0$ km), which is taken to be 18.65 g/kg; and $q_t$ is the specific humidity in the upper atmosphere set to a constant value of $10^{-11}$ g/kg. The constant $z_{q1}$ is set to 4000 m and the constant $z_{q2}$ is set to 7500 m. The analytic initial virtual temperature profile is given by

$$T_v(z) = T_{v0} - \Gamma z \text{ for } 0 \leq z \leq z_t \qquad T_v(z) = T_{vt} \text{ for } z_t < z \qquad (3)$$

where the virtual temperature at the surface $T_{v0} = T_0 (1 + 0.608 q_0)$, the lapse rate $\Gamma$ is 0.0067 Km$^{-1}$, and the virtual tempera-

ture in the upper atmosphere is the constant $T_{vt} = T_{v0} - \Gamma z_t$. $T_0$ is to be set to the SST value for each simulation (295 K, 300 K, or 305 K, respectively). The analytic initial temperature profile $T(z)$ is thus

$$T(z) = \frac{T_v(z)}{1 + 0.608 q(z)}. \qquad (4)$$



The initial pressure profile $p(z)$ is computed using the hydrostatic equation and ideal gas law:

$$p(z) = p_0 \left( \frac{T_{v0} - \Gamma z}{T_{v0}} \right)^{g/(R_d \Gamma)} \tag{5}$$

where $p_0$ is the surface pressure 1014.80 hPa, and $R_d$ and $g$ are given in Table 1. This analytic sounding shown in Fig. 2 is to be used *only* to begin the small domain/single column model simulations, not the large domain/global simulations. It is worth nothing that this analytic setup is similar to that from Reed and Jablonowski (2011) used to initialize tropical environments in GCMs. In addition to being used as spin-up simulations for the large domain/global simulations, the small domain/single column simulations will serve as control simulations that represent "conventional" RCE without convective self-aggregation.

The large domain/global simulations (experiments 1-3) should be initialized with average equilibrium profiles from the small domain/single column simulations (at the corresponding SST). These equilibrium profiles should be derived by taking a horizontal and time mean of the small domain/single column simulation, over the last 30 days of the 100-day simulation (i.e., after the simulation has reached statistical equilibrium). By starting from an equilibrium profile from the more computationally efficient small domain/single column RCE simulations, each large domain/global simulation will begin from that model's own RCE state and thus eliminate the necessity of a lengthy spin-up period with large adjustments. Self-aggregation (which can be thought of as the instability of the RCE state; Emanuel et al., 2014) would be manifest as a large divergence away from the initial state. Care should be taken to ensure the settings of the initial small domain/single column simulations match those of the large domain simulations.

For both the small domain and large domain simulations, symmetry is to be broken by prescribing a small amount of thermal noise in the five lowest layers (an amplitude of 0.1 K in the lowest layer, decreasing linearly to 0.02 K in the fifth layer). This will allow convection to start within the first few hours of each simulation.

## 3.3 Model-type specific settings

The RCE setup described above is to be employed across all models, but we recognize that the domain and numerical details will necessarily be different between CRMs and GCMs, which we describe below. CRMs will employ a limited area planar domain, while GCMs will run on the sphere. We encourage modeling groups (with the capability) to do both RCE on the sphere and on the plane, which will help bridge the gap between CRMs and GCMs. Global cloud resolving models (GCRMs) are a third model type that may participate in *RCEMIP* and represent an important midpoint between CRMs and GCMs. Finally, we also welcome participation from single column models (SCMs), including those not tied to a parent GCM. SCMs should use the configuration described in the previous section.

### 3.3.1 CRMs

For all experiments, cloud resolving model (CRM) simulations, that is, model simulations with explicit convection run on a limited-area planar domain, are to employ a three-dimensional domain with doubly periodic lateral boundary conditions. The small domain simulations used in the initialization procedure are to employ a square domain with 96 grid points in each horizontal dimension with a horizontal resolution of ∼1 km, to approximate the size of a GCM grid box.





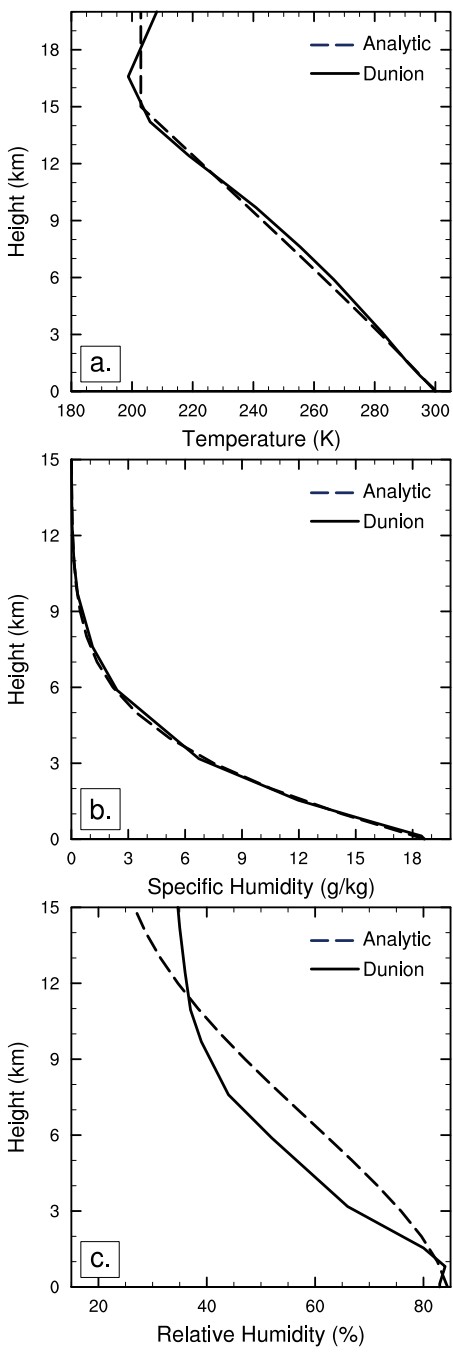

**Figure 2.** Comparison of the analytic vertical profiles to the observed Dunion (2011) moist tropical soundings of (a) temperature, (b) specific humidity and (c) relative humidity (over liquid).





The large domain simulations (experiments 1-3) are to use a grid spacing of ∼3 km, to resolve deep convection and cloud systems but reduce the computational cost. An elongated channel geometry with 2048 grid points in the zonal direction and 128 grid points in the meridional direction (an aspect ratio of 16:1) will allow for the possibility of multiple convectively active regions (if the convection self-aggregates) and the development of large-scale circulations while still simulating three-dimensional convection. Self-aggregation is sensitive to domain size and other numerical details in square geometries (Muller and Held, 2012), but may be more robust in domains with elongated geometries (Wing and Cronin, 2016); this will make comparison across models easier.

The vertical grid will be stretched with at least 64 vertical levels with a model top no lower than 28 km and a sponge layer in the top model layers to damp gravity waves, following a given models usual prescription. Table 3 indicates the recommended vertical grid. The simulations should be run for at least 100 days.

### 3.3.2 GCMs

General circulation models (GCM) that is, models with parameterized convection, should first be run in single column mode to create the equilibrium profile used to initialize the global simulations.

For experiments 1-3, GCM simulations should employ a three-dimension spherical global domain using whichever dynamical core and grid is standard for each given model. Each model should use the horizontal resolution and vertical coordinate and grid of one of their CMIP6 configurations. The simulations should be run for at least 3 years (∼ 1000 days). If the GCM has the capability to run in a planar configuration, it should also be run on the CRM grid described in Sect. 3.3.1, but with the GCM grid spacing and physics parameterizations.

### 3.4 GCRMs

Global cloud resolving models (GCRMs), or "models with explicit convection run on a sphere", are an important category bridging between CRMs and GCMs. Ideally, GCRMs should be run with the same grid spacing as CRMs and the same domain size as GCMs; that is, 3km resolution and the real Earth radius $R_E$, respectively. Although recently more computer resources have become available for running GCRMs at such resolutions, we opt to define a more moderate specification of GCRM experiments such that more research groups running GCRMs are able to join *RCEMIP*. We propose two options: GCRM1: arbitrary horizontal resolution for the sphere with the Earth radius, and GCRM2: 3km horizontal grid spacing for an arbitrary radius of the sphere. Required integration time is the same as that of CRMs (100 days), and the other settings are also the same as CRMs or GCMs, as appropriate.

In practice, relatively coarser resolutions than 3 km are used from GCRMs, though the resolution required to "resolve" clouds explicitly is ambiguous. Resolutions of 7km and 14km are frequently used for NICAM, and even 60km (Yoshizaki et al., 2012) or 220km (Takasuka et al., 2015) have been used without convective parameterizations for capturing MJO-like organized structure (such resolutions without a convective parameterization have also been used in other models, e.g., Becker et al. (2017)). In addition, the definition of the horizontal resolution depends on grid structure and details of discretization which varies among GCRMs, so we recognize that it may not be possible to for all groups to use precisely the same resolution.





If a smaller Earth radius is used, it can be $R_E/2$, $R_E/4$, $R_E/8$, or $R_E/16$ and so on (about 3200 km, 1600 km, 800km, or 400km, respectively). The reduction of the Earth's radius for global RCE studies has also been used in GCMs at hydrostatic scales (Reed and Medeiros, 2016).

To initialize the GCRM simulations, a RCE simulation is first to be performed with the same settings and boundary conditions on a small domain planar version of the GCRM, with a domain size and resolution similar to that used in the small domain CRM initialization simulations (96 km domain length, 1 km horizontal resolution). The average profiles of temperature and humidity from this small simulation should then be used to initialize the global simulation. If a planar configuration of the GCRM is not available, a coarse resolution simulation in which radiation is homogenized to prevent self-aggregation may be used as the initialization simulation. We encourage GCRM groups to contact the *RCEMIP* organizers to discuss appropriate model setups.

## 4 Output Specification

We request output following the conventions of CMIP6 (see http://clipc-services.ceda.ac.uk/dreq/index.html for variable descriptions). The variables should be named according to the CMIP6 names and have the same units as in CMIP6. All variables should be saved for the entirety of experiments 1-3 *and* the small domain/single column simulations used to initialize experiments 1-3. For CRMs, the variables should be output on the model levels and on the native x-/y- grid. For GCMs, the variables should be output on model levels and the native grid (groups may additionally interpolate to the standard CMIP6 pressure levels if they desire). If the native GCM grid is not latitude-longitude, then the output should also be interpolated to a latitude-longitude grid. The output format should be NetCDF, and will be uploaded to a shared data server, which will facilitate analysis and comparison of the simulations.

### 4.1 Variables

Table 4 indicates the list of one-dimensional statistics and domain-averaged profiles that are to be computed and output as hourly averages. The first half of the table is variables that are profiles (function of $z$ and $t$) while the second half is variables that are only a function of time. The italicized variables are non-standard outputs, all others are standard CMIP6 output. The condensed water path, clwvi_avg, includes condensed (liquid + ice) water, and includes precipitating hydrometeors only if the precipitating hydrometeors affect the calculation of radiative transfer in the model. The ice water path, clivi_avg, includes precipitating frozen hydrometeors only if the precipitating hydrometeors affect the calculation of radiative transfer in the model. The vertical coordinate and time coordinate should also be output.

Table 5 indicates the list of two-dimensional variables (functions of $x$, $y$, and $t$) to output, as hourly averages. The italicized variables are non-standard outputs, all others are standard CMIP6 output. The starred variables are outputs for CRMs only. The variables with a $(-)^!$ symbol are outputs for GCMs only. Each model should output one or the other of the variables indicated with a symbol, depending on if they are in height (ˆ) or pressure-based (˜) coordinates. The horizontal coordinates and time coordinate should also be output.





Table 7 indicates the list of three-dimensional variables to output, as instantaneous 6-hourly snapshots. It is optional to upload these variables to the shared data server (we suggest uploading the last 25 days of 3D output), but the 3D variables *must* be saved and stored locally by each modeling group. The italicized variables are non-standard outputs, all others are standard CMIP6 output. The variables with a (-)[!] symbol are outputs for GCMs only. Note that each model should output omega *or* vertical velocity, and geopotential height *or* pressure, depending on whether the model is in pressure-based or height coordinates. Generally CRMs are in height coordinates and GCMs are in a pressure-based coordinate such as hybrid sigma-pressure levels. Each model should output one or the other of the variables indicated with a symbol, depending on if they are in height (ˆ) or pressure-based (~) coordinates). The horizontal, vertical, and time coordinates should also be output.

## 4.2 Diagnostics

### 4.2.1 Cloud Fraction

We request the diagnosis of a global cloud fraction profile that includes all clouds and is the fraction of the entire domain covered by cloud at a given height (it is a function of $z$ and $t$). The presence of a cloud should be defined by an appropriate threshold value of cloud condensate (for CRMs, $1\text{x}10^{-5}$ g/g) or output from cloud parameterizations. This variable should be output along with the other 1D variables (Table 4) under the variable name "cldfrac_avg", for all simulations. For GCMs, we also request the output of a total cloud fraction for each grid column as a 2D variable (Table 5) under the variable name "cl", which is a function of $x$, $y$, and $t$.

### 4.2.2 Aggregation Metrics

We expect that the phenomenon of self-aggregation may occur in some simulations, and therefore plan for the diagnosis of the following aggregation metrics to characterize the degree of aggregation and its spatial scale. There are many other diagnostics that may be computed from the output variables as specified above; here we only list a few key metrics that we expect to be computed from each simulation. Code for each computation will be provided on the RCEMIP website (http://myweb.fsu.edu/awing/rcemip.html).

**1. Organization index** ($I_{\text{org}}$)**:** $I_{\text{org}}$ was introduced by Tompkins and Semie (2017) as an index of aggregation in CRM simulations based on the distribution of nearest neighbor distance between convective entities. First, convective grid cells are defined according to a threshold value of some variable. Then, convective entities are determined by locating connected convective pixels. Next, the distance of the centroid of each convective entity to its nearest neighbor ($d$) is computed. The normalized cumulative distribution function (CDF) of the nearest neighbor distances is then plotted against the theoretical CDF of nearest neighbor distances in a Poisson point process $\left(1 - \exp\left(-\lambda\pi d^2\right)\right)$. The $\lambda$ parameter is estimated as the number of convective entities per unit area in the domain. Finally, $I_{org}$ is defined as the area under the curve plotted in the previous step, and the entire process is repeated throughout the simulation to obtain $I_{\text{org}}$ as a function of time. If the system exhibits random convection behaving as a Poisson point process, $I_{\text{org}}$ would be equal to 0.5. Therefore, values of $I_{\text{org}}$ greater than 0.5 indicate aggregated





convection, with higher values indicating a higher degree of organization. Tompkins and Semie (2017) used a vertical velocity threshold of 1 ms$^{-1}$ at 730 hPa to define updraft grid cells in CRM simulations. Cronin and Wing (2017) compared using a vertical velocity threshold and a cloud top temperature threshold to define $I_{\mathrm{org}}$ and found that, while the absolute values of $I_{\mathrm{org}}$ differed, their tendencies were the same. Therefore, given that *RCEMIP* includes both CRM and GCM simulations and that

a vertical velocity threshold may not be appropriate for GCM simulations, here we define convective grid cells as grid boxes with values of outgoing longwave radiation less than 173 Wm$^{-2}$.

**2. Subsidence fraction (SF):** The degree of aggregation can be characterized by the fractional area of the globe covered by large-scale subsidence in the mid troposphere ($w < 0$ or $\omega > 0$ at 500 hPa), referred to as the subsidence fraction (SF) (Coppin

and Bony, 2015). SF is less than 0.6 when convection is unorganized and greater than 0.6 when it is aggregated. For CRM simulations, the vertical velocity at 500 hPa should be averaged over one day and over a suitably large area ($\sim$100 km, to approximate the size of a GCM grid cell).

**3. Autocorrelation length ($L_{\mathrm{cor}}$):** The autocorrelation length ($L_{\mathrm{cor}}$), was introduced as a method of characterizing the length

scale associated with convective aggregation by Craig and Mack (2013). The autocorrelation function is given by the inverse Fourier transform of the power spectrum of hourly average water vapor path. The autocorrelation length $L_{\mathrm{cor}}$ is the distance at which the autocorrelation function decays to $1/e$ of its maximum value. For a perfectly sinusoidal variable, the wavelength $\lambda$ is related to $L_{\mathrm{cor}}$ by $\lambda = 2\pi[\arccos(\mathrm{e}^{-1})]^{-1}\, L_{\mathrm{cor}} \approx 5.26\, L_{\mathrm{cor}}$; thus, to calculate a length scale representative of the full wavelength of a disturbance, we use the value of $L_{\mathrm{cor}}$ scaled by this factor.

### 4.2.3 Moist Static Energy Budgets

We request that each modeling group estimate the moist static energy budget, as accurately as is possible. Specifically, we request the diagnosis and output of the additional 2D instantaneous variables listed in Table 6. This (along with the other 2D variables) will enable the quantification of the physical mechanisms related to self-aggregation (using the moist static energy variance budget as in Wing and Emanuel (2014)).

Moist static energy is given by $h = c_p T + gz + L_{\mathrm{v}} q$. The values of $c_p$, $g$, and $L_{\mathrm{v}}$ used by the model formulation should be used to compute $h$. The mass-weighted vertical integral of moist static energy (mse) is given by:

$$\widehat{h} = \int_{0}^{z_{\mathrm{top}}} (c_p T + gz + L_{\mathrm{v}} q)\, \rho \, \mathrm{d}z, \tag{6}$$

or, in pressure coordinates,

$$\widetilde{h} = \frac{1}{g} \int_{p_{\mathrm{top}}}^{p_{\mathrm{sfc}}} (c_p T + gz + L_{\mathrm{v}} q)\, \mathrm{d}p. \tag{7}$$



Care should be taken to make sure the same limits of integration are used at all times/locations. The mass-weighted vertical integral of horizontal advective tendency of moist static energy (hadvmse) is given by

$$\int_0^{z_{\text{top}}} \left( u\frac{\partial h}{\partial x} + v\frac{\partial h}{\partial y} \right) \rho \, \mathrm{d}z \tag{8}$$

and the mass-weighted vertical integral of the vertical advective tendency of moist static energy (vadvmse) is given by

$$\int_0^{z_{\text{top}}} w\frac{\partial h}{\partial z} \rho \, \mathrm{d}z. \tag{9}$$

Ideally, moist static energy would be diagnosed online and each model's advection scheme used to advect it, but if this is not possible we ask that groups make their best effort to estimate these terms. The spatial variance of the mass-weighted vertical integral of moist static energy is computed using the squared anomalies from the horizontal mean of the mass-weighted vertical integral of moist static energy $(\widehat{h})$. Its tendency (tnmsevar) is given by

$$\frac{\partial}{\partial t} \left( \int_0^{z_{\text{top}}} h\rho \, \mathrm{d}z \right)^{\prime 2} \tag{10}$$

where $'$ indicates an anomaly from the horizontal mean.

## 5   Sample Results

Several preliminary simulations using the *RCEMIP* configuration have been performed using the System for Atmospheric Modeling (SAM), version 6.8.2 (Khairoutdinov and Randall, 2003), a CRM, NICAM, version 15, a GCRM (Satoh et al., 2014), and the Community Atmosphere Model (CAM), version 5 (Neale et al., 2012), a GCM. We show here sample results from those test simulations as an example of what the *RCEMIP* simulations might look like.

Figure 3-8 show example results from a cloud-resolving model simulation of RCE, using SAM with the settings configured as described in Sect. 3.3.1. Figure 3 shows a snapshot of cloud-top temperature and precipitation rate from the end of a small domain initialization simulation at 300 K; the convection is quasi-random in space and time. Figure 4 shows a snapshot of cloud top temperature and precipitation rate from the end of Experiment 2, a large-domain RCE simulation at 300 K. The convection is aggregated into several large clusters. Self-aggregation is characterized by the development of anomalously moist and dry regions, as can be seen by plots of daily mean water vapor path at day 10 (Fig. 5a) and day 90 (Fig. 5b) of the large-domain SAM simulation. This does not occur in the small domain simulation (Fig. 6); while the domain dries out slightly, the daily mean water vapor path is spatially homogenous. The moist static energy variance budget can be used to diagnose the physical mechanisms contributing to self-aggregation (Fig. 7). The domain average moist static energy variance increases over two orders of magnitude over the course of the simulation, indicating the moistening of moist areas and drying of dry areas (Fig. 7a). Figure 7b shows the contributions of different feedbacks to that growth in moist static energy variance; in this case, it is the surface flux and longwave radiation feedbacks. Finally, we show an example of a metric of aggregation, the subsidence





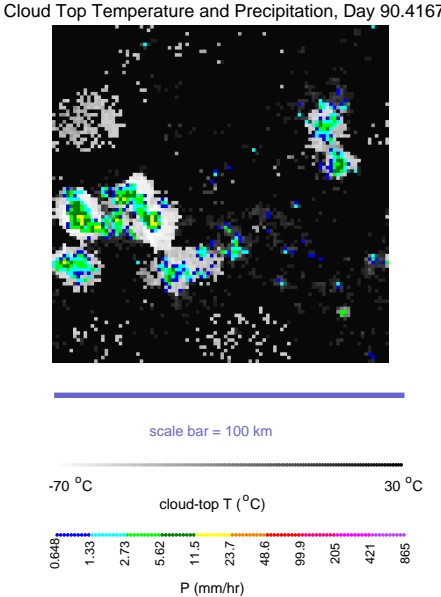

**Figure 3.** Hourly average cloud top temperature (gray shading) and precipitation (color shading) in small domain SAM simulation at $T_s$ = 300K.

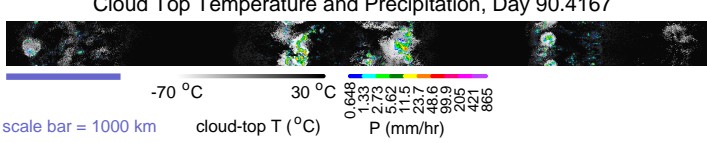

**Figure 4.** Hourly average cloud top temperature (gray shading) and precipitation (color shading) in large domain SAM simulation at $T_s$ = 300K. Note that the scale bar is an order of magnitude larger than in Figure 3.

fraction (SF; Sect. 4.2.2). The subsidence fraction increases over the first $\sim$ 40 days of the simulation, indicating the increasing aggregation of convection and development of large areas of subsiding air (Fig. 8).

Figure 9 shows an example result from a GCRM simulation of RCE, using NICAM in a global, spherical configuration with real Earth radius and a 14-km horizontal grid spacing. Figure 9 shows a snapshot of outgoing longwave radiation (OLR)



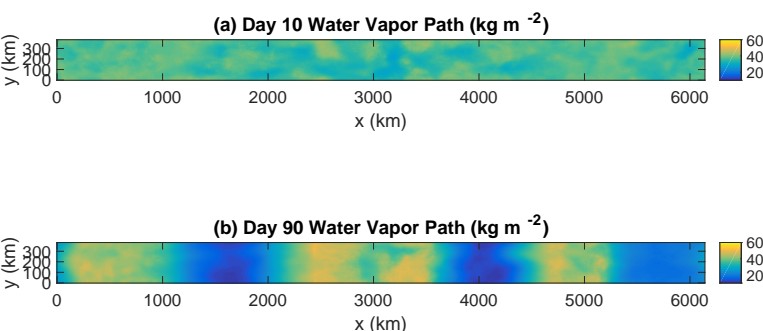

**Figure 5.** Daily mean water vapor path (computed from hourly averages) at day 10 (a) and day 90 (b) of large domain SAM simulation at $T_s$ = 300 K.

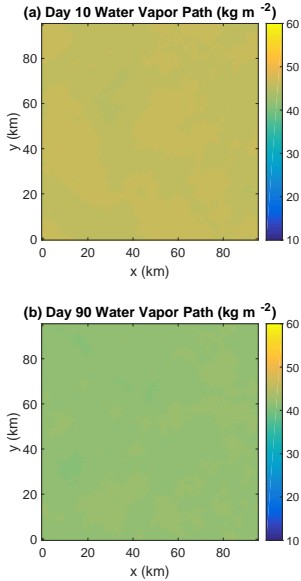

**Figure 6.** Daily mean water vapor path (computed from hourly averages) at day 10 (a) and day 90 (b) of small domain SAM simulation at $T_s$ = 300 K.

and precipitation rate, which is similar to Figs. 3-4. The convection has spontaneously organized into clusters. Differences in aggregation properties, such as cluster sizes, can be seen between the results shown in Fig. 4 and 9, which may stem from the domain geometry, the horizontal resolution, or other details, as mentioned in Sect. 3.3.1. Note that, in this example simulation,



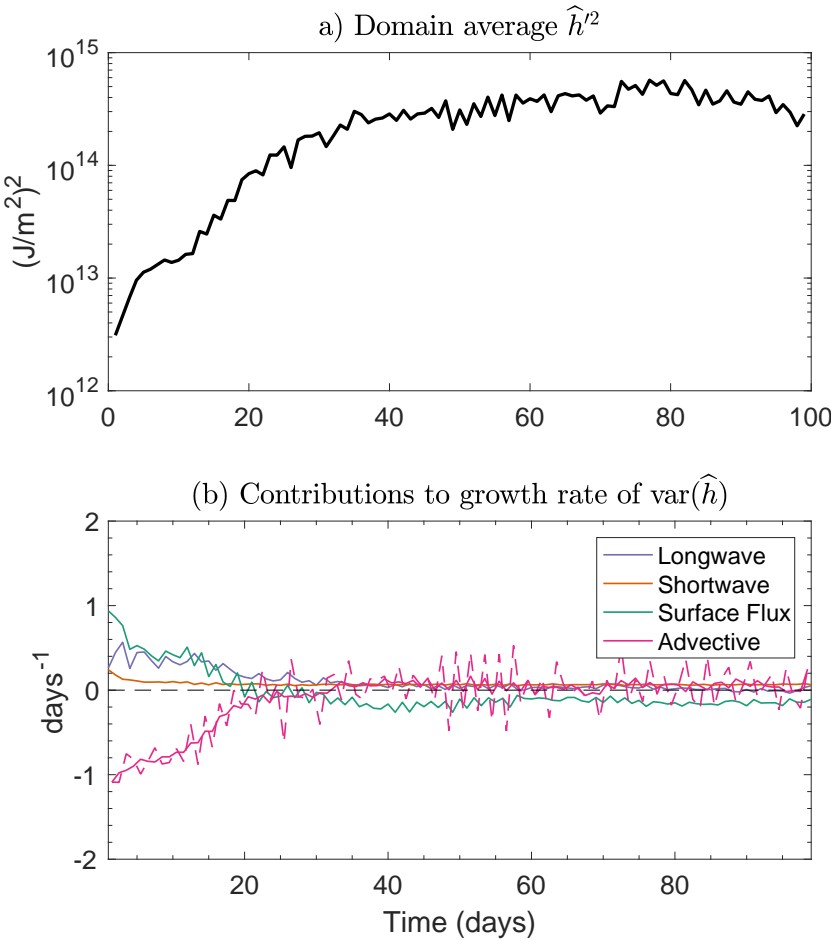

**Figure 7.** Domain average MSE variance (a) and terms in domain average MSE variance budget, normalized by domain average MSE variance (b) from large domain SAM simulation at $T_s$ = 300 K.

the values of 434 Wm$^{-2}$ and 0° were used for the solar constant and the zenith angle, respectively, instead of the values of the *RCEMIP* protocol (Sect. 3.2.2).

Figures 10-12 show example results from a series of GCM simulations of RCE, using CAM5 with the spectral element dynamical core on a cubed–sphere grid with ne30 resolution, which corresponds to ∼100 km grid spacing. More details on the version of CAM5 (including the physics packages) used for these simulations can be found in Reed et al. (2015). Figure 10 shows a snapshot of OLR and precipitation rate for the set of three *RCEMIP* experiments, which can be compared to Figs. 3, 4, and 9. Figure 11 shows a snapshot of water vapor path (at the same time as displayed in Fig. 10). There is a large cluster of clouds and precipitation in each of the simulations at 300 K and 305 K, while the precipitation in the simulation at 295 K is somewhat more scattered. The simulation at 305 K is the most aggregated, with a single hemisphere-scale intensely





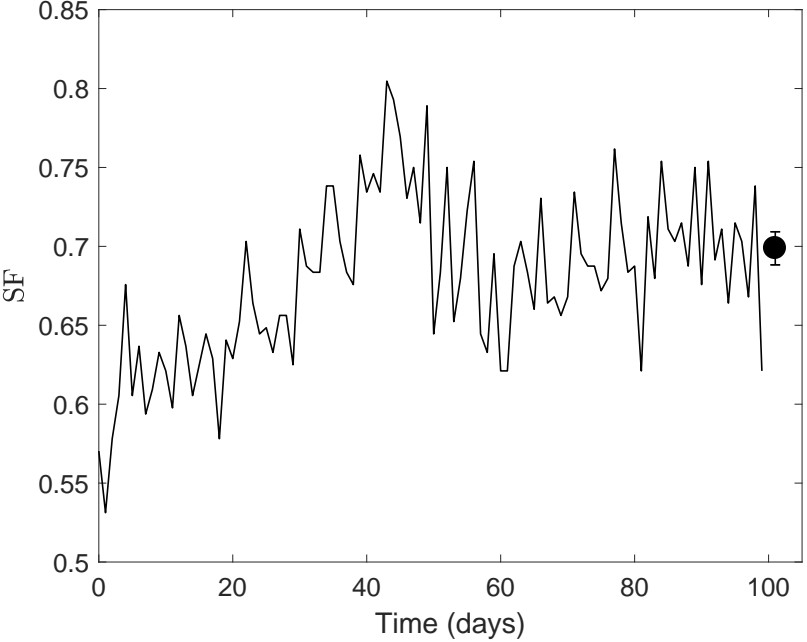

**Figure 8.** Subsidence fraction (SF) as a function of time in the large domain SAM simulation at $T_s$ = 300K. The circle indicates the mean subsidence fraction over the last 25 days of simulation.

precipitating cluster and little cloud cover or precipitation elsewhere on the globe. It is also evident from Fig. 11 that the range of water vapor path values is largest in the simulation at 305 K, with the largest values occurring where the clouds and precipitation are clustered. Figure 12 shows the total global cloud fraction, indicating that anvil clouds shift upward and decrease in amount with warming.

## 6 Participating Models

Table 8 shows a preliminary list of models that are confirmed to participate in *RCEMIP*. We expect this list to grow with participation from other modeling groups and scientists across the world.

## 7 Extensions of *RCEMIP*

We encourage modeling groups to explore other issues based on their interests, which may form the basis for a second phase of *RCEMIP*. There are many additional scientific questions that could be addressed within the framework of *RCEMIP*; here we provide a few suggestions that we think are promising avenues forward but leave open the possibility for other directions that could evolve from the results of the first *RCEMIP* simulations.





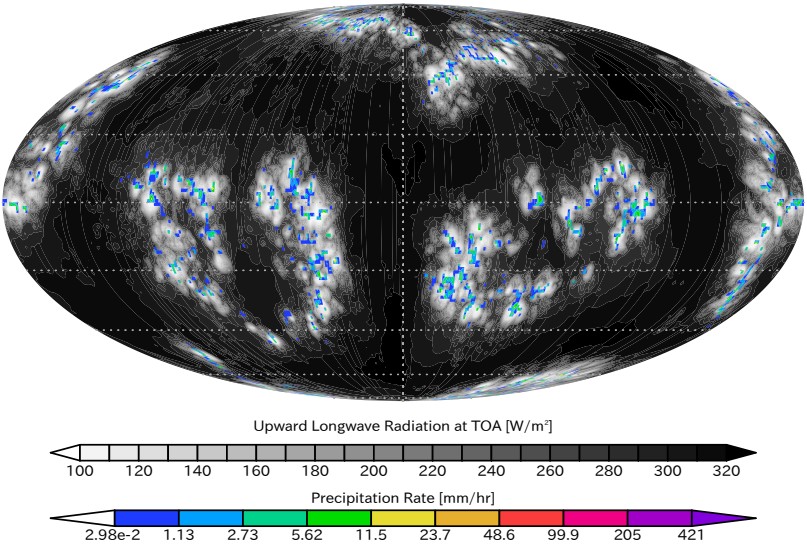

**Figure 9.** Snapshot of outgoing longwave radiation (OLR) at the top of atmosphere (gray shading) and precipitation rate (color shading) in a NICAM simulation at $T_s = 300$ K. Note that, in this simulation, the values of $434\,\mathrm{Wm}^{-2}$ and $0°$ were used for the solar constant and the zenith angle, respectively, instead of the values of RCEMIP protocols.

First, additional simulations could be performed to assess the robustness of the results to model setup/configuration and experimental design. This includes questions about the impact of the lower boundary conditions and dependence on domain size and resolution and initial conditions of convective organization.

Second, additional simulations could be performed to systematically determine the sensitivity of the RCE state and its

changes with warming, and the character of self-aggregation, to the choice of dynamical core and representation of model physics. This includes the sensitivity to dynamical core, radiation scheme, microphysics scheme, convection scheme (in the case of models with parameterized convection), and the sensitivity to various parameters in those schemes (such as the entrainment parameter in a convection scheme). In some cases this could be done within a single model, but *RCEMIP* provides a means of organizing such sensitivity tests across multiple models.

One avenue forward to determine the sensitivity of the RCE state to model setup, dynamics, and physics is to design unified and simple representations of parameterized processes, as for instance was used to study stratocumulus clouds in the GCSS intercomparison study (Bretherton et al., 1999). Such a setup would reduce the ever increasing complexity of parameterizations and thus may be useful for identifying the origin of differences between models. Jeevanjee et al. (2017), in arguing for an "elegant" RCE configuration, suggested that the adoption of a simple, warm-rain, Kessler-type microphysics scheme would

ease comparison between models with regards to cloud fraction and cloud radiative effects, for example.



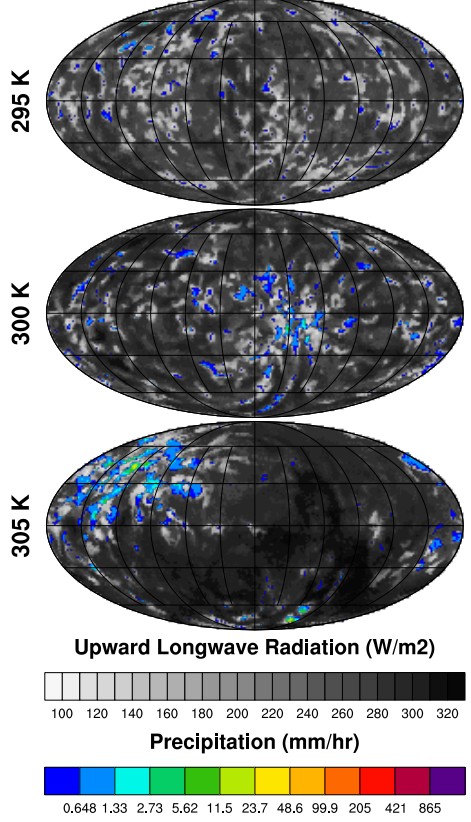

**Figure 10.** Hourly averaged snapshot of upward longwave radiation at the top of atmosphere (OLR; gray shading) and precipitation rate (color shading) from three CAM simulations at $T_s = 295$ K (top), at $T_s = 300$ K (middle), and at $T_s = 305$ K (bottom).

Finally, more in-depth investigation into how the mechanisms of self-aggregation vary across models, including its spatial scale and hysteresis, would be valuable. The initial simulations of *RCEMIP* (Sect. 3) are a good starting point for studying self-aggregation, but further experiments could be defined to leverage the opportunity afforded by *RCEMIP* to make progress on some of the unanswered questions laid out by Wing et al. (2017).

## 8 Conclusions

Radiative-convective equilibrium is an idealization of the tropical atmosphere that, over the past five decades, has led to advances in our understanding of the vertical temperature structure of the tropics, the scaling of the hydrological cycle with warming, climate sensitivity, interactions between convection and radiation, and the development of large-scale circulations. With a coordinated intercomparison including both cloud-resolving models and general circulation models with parameterized convection, *RCEMIP* will help answer important questions surrounding changes in clouds and convective activity with warming, cloud feedbacks and climate sensitivity, and the aggregation of convection and its role in climate. In addition, the simple





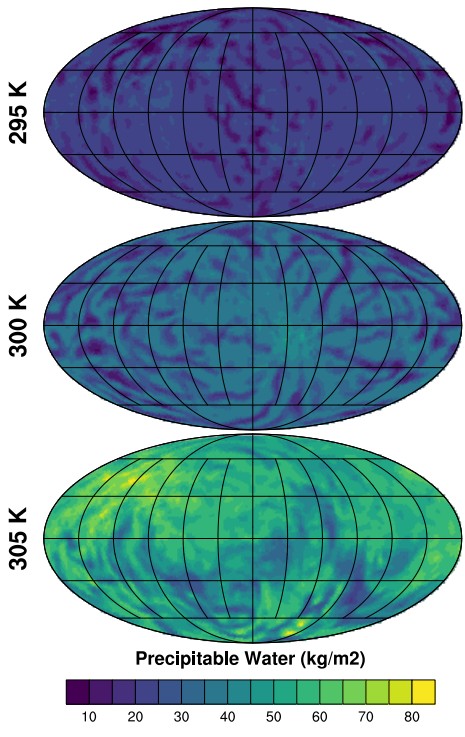

**Figure 11.** Hourly averaged snapshot of water vapor path from three CAM simulations at $T_\mathrm{s}$ = 295 K (top), at $T_\mathrm{s}$ = 300 K (middle), and at $T_\mathrm{s}$ = 305 K (bottom).

premise of RCE will allow the results of *RCEMIP* to be connected to theory, as well as serve as a useful framework for model development and evaluation. *RCEMIP* distinguishes itself from many other intercomparisons because of its ability to involve many model types (SCMs, CRMs, GCRMs, GCMs); the comparison between model types is vital as increasingly higher resolutions are possible in climate-scale global modeling. *RCEMIP* is specifically designed to determine how models of different types represent the same phenomena, and thus provides a basis for testing models with parameterized convection against models that simulate convection directly. In doing so, *RCEMIP* will help us answer some of the most important questions in climate science.

## 9 Code and Data availability

Scripts to calculate the diagnostics described in Sect. 4.2 will be available on the *RCEMIP* website at http://myweb.fsu.edu/awing/rcemip.html. The model output from *RCEMIP* will be made publicly available through the WDCC/CERA archive at DKRZ, accessible at https://cera-www.dkrz.de/WDCC/ui/cerasearch/.



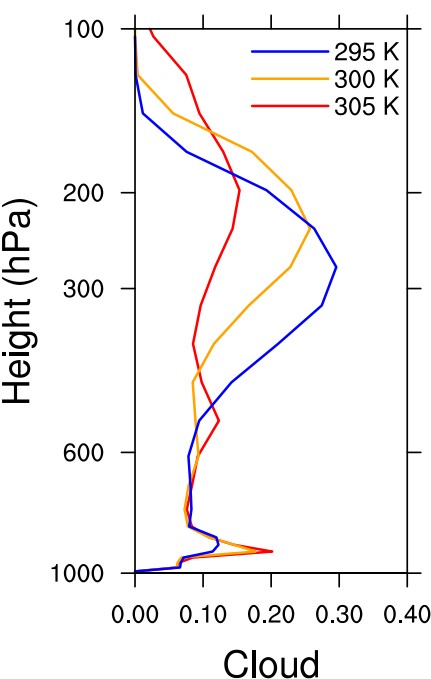

**Figure 12.** Profiles of total global cloud fraction from three CAM simulations at $T_s$ = 295 K (blue), at $T_s$ = 300 K (yellow), and at $T_s$ = 305 K (red).

*Author contributions.* AAW led the writing of the text. All authors contributed to editing the text and discussing the goals and specifications of *RCEMIP*.

*Competing interests.* The authors declare that they have no conflict of interest.

*Acknowledgements. RCEMIP* arose from discussion at the Model Hierarchies Workshop, sponsored by the Working Group on Climate
5   Modeling of the World Climate Research Programme and held in Princeton, NJ in November 2016. The authors thank Nadir Jeevanjee and
Timothy Cronin for helpful feedback and discussion. The System for Atmospheric Modeling (SAM) cloud resolving model is maintained
and provided by Marat Khairoutdinov.

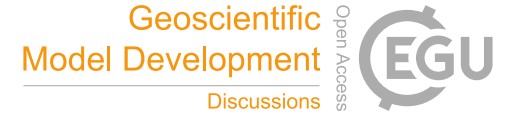

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





**Table 1.** Geophysical constants

| Parameter | Value |
|---|---|
| Earth rotation rate | $\Omega = 0$ |
| Coriolis parameter | $f = 0$ |
| Mean Earth radius | $R_{\mathrm{E}} = 6371.0$ km |
| Mean surface gravity | $g = 9.79764$ ms$^{-2}$ |
| Gas constant for dry air | $R_{\mathrm{d}} = 287.04$ J kg$^{-1}$K$^{-1}$ |
| Specific heat capacity for dry air | $C_{p\mathrm{d}} = 1004.64$ J kg$^{-1}$K$^{-1}$ |
| Water vapor gas constant | $R_{\mathrm{v}} = 461.50$ J kg$^{-1}$K$^{-1}$ |
| Water vapor specific heat capacity | $C_{p\mathrm{v}} = 1846.0$ J kg$^{-1}$K$^{-1}$ |
| Latent heat of vaporization at $0^{\circ}$C | $L_{\mathrm{v}0} = 2.501$ x $10^{6}$ J kg$^{-1}$ |
| Latent heat of fusion at $0^{\circ}$C | $L_{\mathrm{f}0} = 3.337$ x $10^{5}$ J kg$^{-1}$ |
| Latent heat of sublimation at $0^{\circ}$C | $L_{\mathrm{s}0} = 2.834$ x $10^{6}$ J kg$^{-1}$ |



**Table 2.** Radiation and Initialization Parameters

| Parameter | Value |
|---|---|
| *Radiation Parameters* | |
| $CO_2$ concentration | 348 ppmv |
| $CH_4$ concentration | 1650 ppbv |
| $N_2O$ concentration | 306 ppbv |
| CFC11 concentration | 0 |
| CFC12 concentration | 0 |
| CFC22 concentration | 0 |
| CCL4 concentration | 0 |
| $O_3$ fit parameter $g_1$ | 3.6478 ppmv hPa$^{-g_2}$ |
| $O_3$ fit parameter $g_2$ | 0.83209 |
| $O_3$ fit parameter $g_3$ | 11.2515 hPa |
| Solar constant | 551.58 W/m$^2$ |
| Zenith angle | 42.05° |
| Surface albedo | 0.07 |
| *Analytic Sounding Parameters* | |
| $z_t$ | 15 km |
| $q_0$ | 18.65 g/kg |
| $q_t$ | $10^{-11}$ g/kg |
| $z_{q1}$ | 4000m |
| $z_{q2}$ | 7500m |
| $\Gamma$ | 0.0067 Km$^{-1}$ |
| $T_0$ | 300 K |





**Table 3.** CRM Vertical Grid for Scalar Variables

| Level | Height (m) | Level | Height (m) | Level | Height (m) |
|-------|-----------|-------|-----------|-------|-----------|
| 1 | 37 | 23 | 7500 | 45 | 18500 |
| 2 | 112 | 24 | 8000 | 46 | 19000 |
| 3 | 194 | 25 | 8500 | 47 | 19500 |
| 4 | 288 | 26 | 9000 | 48 | 20000 |
| 5 | 395 | 27 | 9500 | 49 | 20500 |
| 6 | 520 | 28 | 10000 | 50 | 21000 |
| 7 | 667 | 29 | 10500 | 51 | 21500 |
| 8 | 843 | 30 | 11000 | 52 | 22000 |
| 9 | 1062 | 31 | 11500 | 53 | 22500 |
| 10 | 1331 | 32 | 12000 | 54 | 23000 |
| 11 | 1664 | 33 | 12500 | 55 | 23500 |
| 12 | 2055 | 34 | 13000 | 56 | 24000 |
| 13 | 2505 | 35 | 13500 | 57 | 24500 |
| 14 | 3000 | 36 | 14000 | 58 | 25000 |
| 15 | 3500 | 37 | 14500 | 59 | 25500 |
| 16 | 4000 | 38 | 15000 | 60 | 26000 |
| 17 | 4500 | 39 | 15500 | 61 | 26500 |
| 18 | 5000 | 40 | 16000 | 62 | 27000 |
| 19 | 5500 | 41 | 16500 | 63 | 27500 |
| 20 | 6000 | 42 | 17000 | 64 | 28000 |
| 21 | 6500 | 43 | 17500 | | |
| 22 | 7000 | 44 | 18000 | | |





**Table 4.** 1D hourly-averaged variables ($z$,$t$) or ($t$)

| Variable Name | Description | Units |
|---|---|---|
| ta_avg | domain avg. air temperaure profile | K |
| ua_avg | domain avg. eastward wind profile | m s$^{-1}$ |
| va_avg | domain avg. northward wind profile | m s$^{-1}$ |
| hus_avg | domain avg. specific humidity profile | kg/kg |
| hur_avg | domain avg. relative humidity profile | % |
| clw_avg | domain avg. mass fraction of cloud liquid water profile | kg/kg |
| cli_avg | domain avg. mass fraction of cloud ice profile | kg/kg |
| *plw_avg* | domain avg. mass fraction of precipitating liquid water profile | kg/kg |
| *pli_avg* | domain avg. mass fraction of precipitating ice profile | kg/kg |
| theta_avg | domain avg. potential temperature profile | K |
| thetae_avg | domain avg. equivalent potential temperature profile | K |
| tntrs_avg | domain avg. shortwave radiative heating rate profile | K s$^{-1}$ |
| tntrl_avg | domain avg. longwave radiative heating rate profile | K s$^{-1}$ |
| tntrscs_avg | domain avg. shortwave radiative heating rate profile - clear sky | K s$^{-1}$ |
| tntrlcs_avg | domain avg. longwave radiative heating rate profile - clear sky | K s$^{-1}$ |
| *cldfrac_avg* | global cloud fraction profile | % |
| pr_avg | domain avg. suface precipitation rate | kg m$^{-2}$ s$^{-1}$ |
| hfls_avg | domain avg. surface upward latent heat flux | W m$^{-2}$ |
| hfss_avg | domain avg. surface upward sensible heat flux | W m$^{-2}$ |
| prw_avg | domain avg. water vapor path | kg m$^{-2}$ |
| clwvi_avg | domain avg. condensed water path | kg m$^{-2}$ |
| clivi_avg | domain avg. ice water path | kg m$^{-2}$ |
| *spwr_avg* | domain avg. saturated water vapor path | kg m$^{-2}$ |
| rlds_avg | domain avg. surface downwelling longwave flux | W m$^{-2}$ |
| rlus_avg | domain avg. surface upwelling longwave flux | W m$^{-2}$ |
| rsds_avg | domain avg. surface downwelling shortwave flux | W m$^{-2}$ |
| rsus_avg | domain avg. surface upwelling shortwave flux | W m$^{-2}$ |
| rsdscs_avg | domain avg. surface downwelling shortwave flux - clear sky | W m$^{-2}$ |
| rsuscs_avg | domain avg. surface upwelling shortwave flux - clear sky | W m$^{-2}$ |
| rldscs_avg | domain avg. surface downwelling longwave flux - clear sky | W m$^{-2}$ |
| rluscs_avg | domain avg. surface upwelling longwave flux - clear sky | W m$^{-2}$ |
| rsdt_avg | domain avg. TOA incoming shortwave flux | W m$^{-2}$ |
| rsut_avg | domain avg. TOA outgoing shortwave flux | W m$^{-2}$ |
| rlut_avg | domain avg. TOA outgoing longwave flux | W m$^{-2}$ |
| rsutcs_avg | domain avg. TOA outgoing shortwave flux - clear sky | W m$^{-2}$ |
| rlutcs_avg | domain avg. TOA outgoing longwave flux -clear sky | W m$^{-2}$ |



**Table 5.** 2D hourly averaged variables ($x,y,t$)

| Variable Name | Description | Units |
| --- | --- | --- |
| pr | surface precipitation rate | kg m$^{-2}$ s$^{-1}$ |
| evspsbl | evaporation flux | kg m$^{-2}$ s$^{-1}$ |
| hfls | surface upward latent heat flux | W m$^{-2}$ |
| hfss | surface upward sensible heat flux | W m$^{-2}$ |
| rlds | surface downwelling longwave flux | W m$^{-2}$ |
| rlus | surface upwelling longwave flux | W m$^{-2}$ |
| rsds | surface downwelling shortwave flux | W m$^{-2}$ |
| rsus | surface upwelling shortwave flux | W m$^{-2}$ |
| rsdscs | surface downwelling shortwave flux - clear sky | W m$^{-2}$ |
| rsuscs | surface upwelling shortwave flux - clear sky | W m$^{-2}$ |
| rldscs | surface downwelling longwave flux - clear sky | W m$^{-2}$ |
| rluscs | surface upwelling longwave flux - clear sky | W m$^{-2}$ |
| rsdt | TOA incoming shortwave flux | W m$^{-2}$ |
| rsut | TOA outgoing shortwave flux | W m$^{-2}$ |
| rlut | TOA outgoing longwave flux | W m$^{-2}$ |
| rsutcs | TOA outgoing shortwave flux - clear sky | W m$^{-2}$ |
| rlutcs | TOA outgoing longwave flux -clear sky | W m$^{-2}$ |
| prw | water vapor path | kg m$^{-2}$ |
| clwvi | condensed water path | kg m$^{-2}$ |
| clivi | ice water path | kg m$^{-2}$ |
| psl | sea level pressure | Pa |
| tas | 2m air temperature | K |
| *tabot** | air temperature at lowest model level | K |
| uas | 10m eastward wind | m s$^{-1}$ |
| vas | 10m northward wind | m s$^{-1}$ |
| *uabot** | eastward wind at lowest model level | m s$^{-1}$ |
| *vabot** | northward wind at lowest model level | m s$^{-1}$ |
| *wa500^* | vertical velocity at 500 hPa | m s$^{-1}$ |
| *wap500~* | omega at 500 hPa | Pa s$^{-1}$ |
| *spwr* | saturated water vapor path | kg m$^{-2}$ |
| cl$^!$ | total cloud fraction of grid column | % |



**Table 6.** 2D instantaneous hourly variables ($x$,$y$,$t$)

| Variable Name | Description | Units |
|---|---|---|
| mse | mass-weighted vertical integral of moist static energy | J m$^{-2}$ |
| hadvmse | mass-weighted vertical integral of horizontal advective tendency of moist static energy | J m$^{-2}$ s$^{-1}$ |
| vadvmse | mass-weighted vertical integral of vertical advective tendency of moist static energy | J m$^{-2}$ s$^{-1}$ |
| tnmse | total tendency of mass-weighted vertical integral of moist static energy | J m$^{-2}$ s$^{-1}$ |
| tnmsevar | total tendency of spatial variance of mass-weighted vertical integral of moist static energy | J$^2$ m$^{-4}$ s$^{-1}$ |

**Table 7.** 3D instantaneous hourly variables ($x$,$y$,$z$,$t$)

| Variable Name | Description | Units |
|---|---|---|
| clw | mass fraction of cloud liquid water | g/g |
| cli | mass fraction of cloud ice | g/g |
| *plw* | mass fraction of precipitating liquid water | g/g |
| *pli* | mass fraction of precipitating ice | g/g |
| mc$^!$ | convective mass flux | kg m$^{-2}$ s$^{-1}$ |
| ta | air temperature | K |
| ua | eastward wind | m s$^{-1}$ |
| va | northward wind | m s$^{-1}$ |
| hus | specific humidity | g/g |
| hur | relative humidity | % |
| wap$^{\sim}$ | omega | Pa s$^{-1}$ |
| *wa*$^{\wedge}$ | vertical velocity | m s$^{-1}$ |
| zg$^{\sim}$ | geopotential height | m |
| *pa*$^{\wedge}$ | pressure | Pa |
| tntr | tendency of air temperature due to radiative heating | K s$^{-1}$ |
| tntc$^!$ | tendency of air temperature due to moist convection | K s$^{-1}$ |
| *tntrs* | tendency of air temperature due to shortwave radiative heating | K s$^{-1}$ |
| *tntrl* | tendency of air temperature due to longwave radiative heating | K s$^{-1}$ |



**Table 8.** Preliminary List of Participating Models

| Model | Acronym | Model Type | Citation |
|---|---|---|---|
| Community Atmosphere Model, version 5 | CAM5 | GCM | Neale et al. (2012) |
| Community Atmosphere Model, version 6 | CAM6 | GCM | TBD |
| ECHAM6 | ECHAM6 | GCM | Popke et al. (2013) |
| ICOsahedral Nonhydrostatic Model | ICON | CRM/GCRM/GCM | Dipankar et al. (2015) |
| IPSL-CM5A-LR | IPSL-CM5A-LR | GCM | Dufresne et al. (2013) |
| IPSL-CM6 | IPSL-CM6 | GCM | TBD |
| Nonhydrostatic ICosahedral Atmospheric Model, version 15 | NICAM.15 | GCRM | Satoh et al. (2014) |
| System for Atmospheric Modeling | SAM | CRM | Khairoutdinov and Randall (2003) |
| UCLA Large-Eddy Simulation Model | UCLA-LES | CRM | Hohenegger and Stevens (2016) |