# Peer review of "Radiative-Convective Equilibrium Model Intercomparison Project"

_Geoscientific Model Development, 2017_

## Referee Comment (RC1) · I. Held (Referee) · 24 Sep 2017

I am very supportive of the proposed RCE intercomparison project. The paper is generally well-written, and my comments primarily relate to the specifics of the proposed project.

Major issues:

I continue to be puzzled that the papers on RCE simulations with CRMs hardly ever discuss the sensitivity of the results to boundary layer formulations. In fact, most of these simulations have no boundary layer closure at all except for the Monin-Obukhov (M-O) surface flux framework. M-O-is, in fact, a boundary layer formulation, and if that is all that one has in the model, one is (arbitrarily it seems to me) assuming that a

closure is needed between the surface and the lowest model layer but not between the first model layer and the second model layer. The specification of a minimum wind speed in the bulk surface flux formula can also be thought of as a boundary layer turbulence parameterization of sorts, and the fact that you are careful to include this in your specification is implying that you don't trust the resolved scales to provide the right gustiness. Running at 3km resolution without a boundary layer above the lowest model level strikes me as extreme (not that I haven't done it myself). I would really like to see this topic directly discussed by the authors. This is a "grey zone" that seems to get less attention than the problem of transitioning from parameterized "convection" to explicit "convection" and one that I find very confusing.

Along the same lines, it would be nice if the text more explicitly encouraged much higher resolution simulations than 1km in the 100km domain, closer to LES resolution for the planetary boundary layer.

I would strongly encourage you to reconsider and ask groups to run with a standard microphysical mechanism in addition to their model's microphysics. Otherwise, there is a good chance that the diversity of simulations will be dominated by the diversity of microphysical assumptions. (For example, we know that in RCEs assumptions about ice fall-speeds will exert a strong control on the cirrus climate.) I realize that this would be work for the authors, who have to be confident that this standard microphysical mechanism worked reasonably well in a couple of the models that they are familiar with. The proposal here is to leave this issue to a second round, but I think this is postponing the inevitable. (For example, you suggest that models with more elaborate microphysics, with explicit activation of CCNs perhaps, be adjusted to create simulations that produce the same droplet number concentrations as recommended for models that specify these droplet numbers, but what if droplet numbers change significantly with warming in these models?)

The utility of single column models, in my view, is dependent on their "effective stability": the single column equilibrium, whether time-independent or dependent, with no

resolved flow is always a solution of a fully doubly periodic box or global RCE model that uses the same column physics (in fact, this is a good test that a global model has been successfully homogenized) . But it is invariably spatially unstable. To the extent that the time average of the RCE state in the 100km box bears little resemblance to the single column equilibrium, the single column results are going to be hard to relate to the rest of the project.. I would try to encourage any contributions of single column RCE simulations to also provide full doubly-periodic 100km box simulations for comparison, or at least global RCE simulations if these are easier to perform.

For the same reason, I don't particularly care for initializing global models with single column equilibria. Among other issues, what if these equilibria are time-dependent? (This is not hypothetical – it may be the rule rather than the exception with cloud-radiative interactions present). Do you initialize all the grid point with the same phase?

I would not specify number of grid points to be used. I would just specify the physical size of the domain and then encourage simulations with 1km resolution in the small domain and 3km in the bowling alley domain, while also encouraging any other resolutions that the groups are interested in exploring.

The recommended upper boundary seems quite low to me, making it awkward to look at the vertically propagating gravity waves in the stratosphere excited by the convection. This is a nice opportunity to look at the robustness of the simulation of those waves, especially in the bowling alley case,. On a related point, I would be a little worried that the bowling alley configuration will in some cases allow the generation of QBO-like oscillations, as in the two-dimensional limit. This could cause some large unexpected differences between simulations. It is worth mentioning at least.

You seem to be initializing stratospheric water with extremely small, but non-zero, mixing ratios. But won't stratospheric water take forever to equilibrate in RCE simulations, where the only thing that can mix vapor up is breaking gravity waves? This vapor has some effect on sensitivity of radiative fluxes to warming. Kuang and Bretherton, JAS,

2004 have circumvented this problem by specifying a small vertical motion (a Brewer-Dobson upwelling extending through the troposphere and stratosphere), which allows the tropopause cold-trap to operate. As in nature This also makes it easier to study the tropical tropopause layer, where convection and upwelling can both be important . In any case, this issue of equilibrating stratospheric water should be discussed. (How do you get water vapor into the stratosphere in a single column model? Do you intend that these models include sub-grid closures for mixing by convectively generated waves?)

Minor comments:

p-3, l-2. Held, Zhou, Wyman, 2007 also compared the cloud feedbacks in RCE against those in the tropics of a global model with the same column physics.

p.-5, l-20. Sorry, but false precision is a pet peeve of mine. You specify the mean surface pressure to 6 significant figures, but the mass of the atmosphere is not conserved, it increase with warming due to increase in vapor. Some global models correctly incorporate the mass source/sink of vapor, and these models will just ignore the value that you specify, they need to specify the dry mass. As another example, in the list of constants you are careful to specify the value of the latent heat of vaporization , but you don't specify its temperature dependence – or, equivalently, the heat capacity of the condensate. Are you expecting the latter to be set to zero? You specify the (mean) radius of the Earth with precision and then suggest that models just use smaller radii, a good idea since the radius in the non-rotting case is totally irrelevant until it approaches the depth of the model domain. Meanwhile differences in assumptions about surface smoothness or ice fall-velocities or many other things that are literally orders of magnitude more important are passed over. The table of geophysical constants is pointless in my opinion; just ask modelers to use their standard Earth values

I would not ask the modeling group to provide your preferred measures of aggregation. Other measures might turn out to be more interesting. These should be relatively easy to generate in a uniform way after the fact from the data archive. You should keep the

post-processing requested from the groups to a minimum. (Some groups may not be interested in the dynamics of aggregation.)

As for the energy budget, I am not sure why you want this in advective rather than flux form (I may be missing something.). A data request could include the fluxes of energy at the boundaries of the grid cells – at least in CRMs and grid-point GCMs. These are more likely to be precisely defined by the underlying numerical scheme than the advective form.

I can't see anything in Fig. 6

––––––––––––––––––––––––––––––––

---

## Referee Comment (RC2) · L. Silvers (Referee) · 8 Oct 2017

General Comments:

An intercomparison of models in RCE is an excellent idea. This paper provides a good framework to help organize the project and build on the current momentum that the topic of RCE has. The paper is well written.

The current level of organization for this project is impressive and will hopefully lead to many participating models. The availability of diagnostic codes is also a major benefit.

I think the third theme on robustness is one of the most critical. It gets a bit lost in the discussion about aggregation and sensitivity to warming. When the literature on RCE is surveyed, the apparent diversity of results is often the primary first impression. For example, not all RCE models aggregate. The WRF model in RCE mode (for a particular configuration) does not aggregate while SAM does (see last chapter of dissertation by Wenyu Zhou). The DAM model does not spontaneously aggregate (Jeevanjee and Romps, 2013). The broad range of results displayed in Held, Zhao, and Wyman, 2007 illustrates the difficulty of establishing a 'ground-truth' or baseline for RCE experiments. Some RCE models generate a 'QBO-like' oscillation, and some have an issue with dominant and persistent upper-level clouds. Another example of large variations in RCE results is shown in Silvers et al. 2016 where the ICON model exhibits large differences in the climate sensitivity for differing domain sizes despite similar subsidence fractions. There are consistent and robust results from RCE models, and I think much can be learned from them, establishing the bounds within which RCE is consistent among models and without which RCE varies across configurations would be one of the most significant/useful scientific outcomes of RCEMIP. The authors are clearly aware of this general point (lines 16-18 of page 2), but it could be clearer in the text, and emphasizing the importance of this component could make the project more appealing to some potential participants.

It will be very useful to the community to determine the response of clouds to warming, but this will have to be interpreted through the lens of RCE and the applicability to observations will be dependent on our ability to establish a consistent picture of the RCE results. This is why I think the current first theme, clouds and climate sensitivity, is secondary to the robustness theme.

It is not essential, but I think it would be useful to be more precise about a second set of non-required (Tier 2) experiments. This could be written in such a way that modeling centers wishing to participate with minimal effort are not thus discouraged from participating, but that more ambitious modeling centers or individuals could clearly push farther into the project in a coordinated way. My suggestions for further experiments would be:

1. Rotating RCE 2. GCMs in RCE mode with convective parameterization turned off

3. RCE with cloud RCE off (COOKIE type experiments) 4. Kessler physics across the hierarchy of models

This doesn't necessarily need to be incorporated in the paper: Presenting the hierarchy of model types as having 3 tiers misses a critical tier of a GCM with simplified microphysics. If we are discussing tiers and hierarchies why not state five model types for RCE: LES-RCE (sub km) CRM-RCE (1-5km), DoublyPeriodic_CourseRes-RCE(simple physics), DoublyPeriodic_CourseRes-RCE(full physics), and a global GCM-RCE(full physics).

Figures:

Overall the figures are clear, well formatted, and useful. However, the current set of figures isolates the different models rather than capitalizing on the intercomparison.

It would illustrative of the motivation for RCEMIP to have a figure illustrating the same quantity across the hierarchy of model configurations. For example, it would be nice to see something like a plot of vertical mean cloud fraction from SAM, NICAM, and CAM; a panel plot showing OLR (or water vapor path) from these three model configurations; or a panel plot showing the subsidence fraction as a function of time for the three model types.

Figure 9: It isn't a big deal, but why weren't the RCEMIP protocol parameters used for this figure?

Figures 12-14: I am assuming that these figures are showing data from the same 3 CAM simulations. Am I right? If so, this is not explicitly mentioned in the captions, and should be.

Specific Comments:

Page 2:

Lines 16-18: This seems to say that the reason it is still unclear if the observed atmosphere aggregates is because details of aggregation in models are dependent on model formulation. But in my opinion the lack of clarity on aggregation in observations and its relevance to climate change is mostly due to the difficulty of connecting an RCE model to cases with strong dynamics and horizontal gradients of forcing.

Line 27: I agree that it will be useful to extend the range of models used to simulate RCE, but single-column models have already been used to study, or in coordination with, RCE. So single-column models don't extend the range that already exists, nor will the proposed experiments here. What RCEMIP will do in this regard is to help fill-out the ensemble space of a baseline set of model configurations. This is important to evaluate the generality ('robustness') of the previous work on RCE, and will help to identify or map out what types of RCE models are still needed in the hierarchy to better our understanding.

Line 30: Citing Silvers et al. here is questionable, as we did not make the comparison suggested, rather out intention was to motivate the comparison. All of our domain sizes had the same grid-spacing. Page 2, line 2 would be more appropriate. Also page 4, line 32 (on comparing the spread of climate sensitivity) would work, but is not necessary.

Concerning the themes outlined, themes 2 and 3 seem quite broad and as a result a bit vague. Perhaps broadness is the aim?

Lines 2-4: This sentence seems to be overly emphasizing what RCE would tell us about clouds in the observed atmospheric system. The connection between the climate sensitivity of RCE simulations and the fundamental characteristics of modeled clouds is almost clear, but what are the fundamental characteristics of modeled clouds, perhaps you could say the climate sensitivity is one of those characteristics but not all encompassing? My interpretation is that the RCE convection is a key baseline that will

help us understand the role of modeled convection in determining the climate sensitivity.

It would be useful to ask for the ice-fall speed, and the fraction of convective precipitation from models to be saved. These would be helpful to categorize or interpret seemingly different results (as discussed in Held, Zhao, and Wyman, 2007).

Technical Corrections:

There should be a "to" on line 9 between "guide" and "those who"

Line 14: Perhaps delete this comment? 'mimicking' sounds like you are playing tricks here, or like you are not really representing the state described. I think something like, 'representing' would be more accurate. You are representing the described state. Shortcomings in the representation come from RCE, or the model itself, not from the boundary conditions.

Line 24: 'conditions' should be singular.

Page 10:

Line 28: "from" should be "for", "with", or "in".

Line 33: There is an extra "to" in the last line.

Should more be said about necessary adjustments when altering the radius of earth? Maybe just citing more of Kevin Reed's work would be an easy way to give people a clue about the details without adding too much technical material to the paper.

Page 13:

Line 2: Should "aggregation" be added here to indicate what was compared? As written, it is a bit unclear and the sentence should be reworded a bit for those who

have not read Cronin and Wing.

Page 14:

Line 17: 'Figure' should be plural.

Page 15:

Should we really be calling a GCM with 14 km horizontal grid-spacing a cloud resolving simulation (GCRM)?
* * *

---

## Referee Comment (RC3) · Anonymous Referee #3 · 19 Oct 2017

Overall, I am very supportive of the initiative and recommend publication of this protocol paper in the non-discussion GMD journal. The discussion paper is largely focused on aggregation questions, which are a pressing scientific concern that this intercomparison will be critical to addressing, but there is also a lot of value in the small domain simulations (what does the cloud fraction look like there? e.g., Fig. 12). So, some additional discussion of the value of the MIP independent of aggregation questions would be well justified. I look forward to seeing the science enabled by RCEMIP.

Major points:

(i) I found it odd that the presentation of the preliminary results of the intercomparison were separated by model. Why not actually compare, for example, the OLR of the simulations in the same figure with the same color bar (vs. Fig. 3, 9, 10)? Even if

the authors decide to leave the separate structure (SAM then NICAM then CAM5), the figures should allow for an "apples-to-apples"comparison (precipitation rate intervals and colors differ between Fig. 9 and 10, for instance). Last, Fig. 12 should either have analogous results from other models or not be included.

(ii) The analytic initial condition for the small domain simulation needs to be motivated. It's only necessary if multiple equilibria are simulated. Otherwise, it's an additional barrier to participation.

(iii) Description of domains: p.10: Is it correct that only a rectangular channel geometry is part of the MIP for CRMs? This is a non-obvious choice, so it would be good to state clearly that there is no square domain simulation requested. Do you think it's best to specify approximate horizontal resolution and a fixed number of grid points in the two directions? I would prefer the opposite: like Lx = 6000km, Ly = 400km. Also, GCRMs are still non-rotating? I wasn't sure how to interpret L20 says "run on a sphere". This is like the CAM simulations shown: non-rotating but Earth's geometry, right?

(iv) Aggregation metrics: I was unsure about the need for some of these to be pre-computed by the participants. The Organization index proposed is determined by the OLR distribution, so that's something that isn't necessary to pre-compute. Likewise for the subsidence fraction, unless there is something about the temporal frequency of the output that I missed. In contrast, I definitely understand that it's valuable if the modeler/modelling center provides moist static energy budgets.

(v) Climate sensitivity and connection to more comprehensively configured GCMs: Perturbing SST allows for a quantification of the "Cess-sensitivity", but the 5 K interval for the SST perturbation is larger than uniform warming simulations in comprehensive MIPs. So, please provide additional motivation for this choice. I believe it may also be "big" from the aggregation perspective (Wing et al 2014 had big changes from unaggregated to near-peak aggregation for a comparable magnitude perturbation). The other aspect of climate sensitivity that this kind of intercomparison (with both GCMs and

CRMs) could address is tropospheric cloud adjustments to changing CO2. It would be good to see how GCMs compare to CRMs in this aspect of their sensitivity (see, e.g., Wyant et al. 2012 JAMES doi:10.1029/2011MS000092). It's also an important complement to narrowly comparing the Cess sensitivity between RCE and Earth boundary conditions. The RCE configuration may have a different radiative forcing because of the differences in the control simulation cloud distribution or differences in cloud adjustments. For example, compared to GCMs, RCE may have a big Cess-sensitivity (in the large warming sense), but also a smaller radiative forcing.

Minor comments:

* author affiliations out of order 4 and 5

* p. 1 L4 "role of self-agg..." on what?

* single column models -> single-column models (though maybe conventional compound-adjective practices aren't used for SCM)

* p.1 L15 climate sensitivity estimates: expect a Manabe and Weatheral 1967 reference here

* p. 2 L9 a couple of earlier RCE precip extremes papers from Muller, Romps

* p. 2 "formulaic sensitivity" I found this confusing

* p. 6 L 29 "the lapse rate" -> " the virtual temperature lapse rate"

* p. 8 after going through the description of the continuous analytic formula for the initial condition, a discrete near-surface perturbation is used (lowest 5 layers); the perturbation seems irrelevant to GCMs, unless I'm missing something.

* p. 10 L 12 missing punctuation after GCM

* p.11 L21 stray )

* p.12 Do you want to specify the variable names for the horizontal coordinates of the

CRM output? If participants are forced to convert variables names to the CMOR format, it would be good also do something consistent for the coordinates (I'm not suggesting labelling them latitude and longitude, to be clear).

* p.18 Fig 8 caption: what are whiskers? standard deviation? Some of the simpler aggregation metrics should be evaluated in the non-SAM simulations (see major issue 1)

─────────────────────────────

---

## Referee Comment (RC4) · Anonymous Referee #4 · 19 Oct 2017

The authors suggest an intercomparison project for various types of models run in radiative-convective equilibrium (RCE). While previous studies have differed in details of their setup the aim of this paper is to provide a common baseline. First, the setup of the intercomparison is detailed, then some sample results are provided.

Overall, I think that this is a great initiative and the suggested setup useful. There are a few suggestions to consider:

1. As e.g. detailed in Wing et al. (2017), the resulting equilibrium state and the clustering may look very different in the different CRMs. It will thus be very difficult to compare the different models and to identify the root for the differences (radiation scheme, microphysics parametrizations, . . .). An even simpler setup for the models could therefore be useful to identify, which schemes are responsible for the differences. As suggested

by the authors and brought forward by Jeevanjee et al. (2017) a simplified microphysics scheme could be one option. A further option could be to simplify the longwave radiative cooling, as e.g. described in Muller and Bony (2015). Such a simplified simulation should be run as number 0 at one SST to assess science objective 3. If this simulation already showed large differences between the individual CRMs running simulations 1-3 should be reconsidered.

2. A number that is hardly discussed in the aggregation literature is the heating/cooling rate at which the equilibrium is reached. Which longwave cooling rate is balanced by convection? How much latent heat is released? This will again be reflected in the surface precipitation rate. How strongly does this number differ between simulations and what is its value in observations? The output from multiple models would give an indication of how much this value varies, and how GCMs with parametrized convection compare with CRMs/observations. A value of $\sim$100 W/m2 for precipitation is found, but how much variation around this value is there between the participating models? On page 4, last paragraph it is mentioned that radiative flux divergence is nearly a universal function of temperature, which in turn is a function of temperature only. Before investigating the response of RCE to warming it is important to first focus on the robustness of these fundamental quantities across models. 3. A strong focus of the project will lie on the coupling between convection and circulation. At the same time it is stressed that the proposed framework will be suitable for SCMs, that are unable to resolve circulations by design. Moreover, aggregation can not occur. Please specify more clearly which aspects from the SCMs will be analyzed and how they can contribute to a deeper understanding of aggregation in RCE.

Specific comments:

page 2, line 14: correct "explict" page 3, line 4: remove "in" before "between" page 3, line 9: add "to" before "those" page 10, last paragraph: a grid spacing of 60 or even 220 km very likely not produce reasonable convection, which makes the acquired RCE state questionnable. page 10, line 33: remove "to" page 13, line 21: replace "estimate"

with "estimates" page 20, line 4: please list the unanswered questions laid out by Wing et al., (2017) for those readers who are not very familiar with the paper. Figures 9 and 10: please give the point in time
* * *

---

## Editor Comment (EC1) · C. van Heerwaarden (Editor) · 30 Oct 2017

In my view, all reviewers clearly see the benefit of the proposed project. Each of the reviews is of high quality and provides clear suggestions on how to improve the manuscript and the intercomparison project. I suggest that you reply to those and proceed with preparing a revised manuscript.

---

## Short Comment (SC1) · 8 Nov 2017

Apologies for the late post....

About a half dozen GFDL scientists, along with two GFDL post-docs, met to discuss the RCEMIP proposal and GFDL's potential participation using the FV3 dynamical core. This note outlines some of the thoughts and responses that came up.

There is a common interest here in developing a doubly-periodic, cloud-resolving RCE configuration of FV3 which uses our current ('AM4') comprehensive physics package and which would be suitable for RCEMIP. Motivations for this are diverse, however. Probably the most common motivation is to use RCE as an idealized testing ground, for use in (say) comparing microphysics schemes, or benchmarking low-resolution

parameterized-convection simulations against high-resolution explicit convection simulations on the same domain. Note that these activities have little to do with RCEMIP as currently presented, though there is some mention (but little emphasis) in the manuscript on planar GCM configurations.

A second motivation for our development of FV3 RCE would be to assess how idiosyncratic our simulated, unaggregated RCE state is relative to other models. This falls nicely in line with the 'robustness' objective of RCEMIP (science objective #3), though (as other reviewers have pointed out) this objective seems to get de-emphasized in the paper. The authors may want to consider increasing their emphasis on it.

Convective self-aggregation seems to be a major focus of RCEMIP, and while there is some interest in aggregation here at GFDL, overall it is probably only a secondary concern. Thus, while interest in a suite of kilometer (or even sub-kilometer) small-domain RCE simulations is strong, interest in aggregated simulations (and especially the computation of secondary, aggregation-focused diagnostics) seems to be weaker.

As advocated for by Isaac and other reviewers, there is also interest here in using simplified (Kessler) microphysics in our RCE setup. Such a scheme already exists in development branches of our code.

---

## Author Comment (AC1) · 18 Dec 2017

See attached.

Please also note the supplement to this comment:
https://www.geosci-model-dev-discuss.net/gmd-2017-213/gmd-2017-213-AC1-supplement.zip

---

## Author Comment (AC2) · 18 Dec 2017

See attached.

Please also note the supplement to this comment: https://www.geosci-model-dev-discuss.net/gmd-2017-213/gmd-2017-213-AC2-supplement.zip
* * *

---

## Author Comment (AC3) · 18 Dec 2017

See attached.

Please also note the supplement to this comment: https://www.geosci-model-dev-discuss.net/gmd-2017-213/gmd-2017-213-AC3-supplement.zip
* * *

---

## Author Comment (AC4) · 18 Dec 2017

The comment was uploaded in the form of a supplement:
https://www.geosci-model-dev-discuss.net/gmd-2017-213/gmd-2017-213-AC4-supplement.zip

---

## Author Comment (AC5) · 18 Dec 2017

See attached.

Please also note the supplement to this comment:
https://www.geosci-model-dev-discuss.net/gmd-2017-213/gmd-2017-213-AC5-supplement.zip
* * *

---

## Author Comment (AC6) · 18 Dec 2017

Dear Editor,

Thank you for the encouragement to revise our manuscript. We have addressed the comments by the 4 reviewers and additional short comment; please see the point-by-point responses that we posted in reply to each. Each response also contains a tracked-changes version of the revised manuscript.

Sincerely,

Allison Wing
* * *
[Figure]

2017.

---

## Author Response (AR1)

I am very supportive of the proposed RCE intercomparison project. The paper is generally well-written, and my comments primarily relate to the specifics of the proposed project.

We thank Dr. Held for his support of RCEMIP and constructive comments regarding the specifications of the proposed simulations. We have made several changes to the details of the simulation specifications in response (see revised manuscript and responses to individual comments below). In addition, we hope that the revised manuscript (see especially Section 6) better conveys our goal with regards to the simulation design: to keep it as simple as possible in order to maximize participation, establish a baseline, and inform subsequent experimentation. We envision that RCEMIP will eventually extend far beyond the simulations laid out in this paper, as we recognize that the baseline we propose will not be a definitive representation of the RCE climate for many of the reasons raised by the reviewers. While it is clear that certain physical parameterizations (or lack thereof) may lead to biases that require further investigation, we see the simulations proposed as a way to bring the community together to get us to that next point.

Major issues:

I continue to be puzzled that the papers on RCE simulations with CRMs hardly ever discuss the sensitivity of the results to boundary layer formulations. In fact, most of these simulations have no boundary layer closure at all except for the Monin-Obukhov (M-O) surface flux framework. M-O-is, in fact, a boundary layer formulation, and if that is all that one has in the model, one is (arbitrarily it seems to me) assuming that a closure is needed between the surface and the lowest model layer but not between the first model layer and the second model layer. The specification of a minimum wind speed in the bulk surface flux formula can also be thought of as a boundary layer turbulence parameterization of sorts, and the fact that you are careful to include this in your specification is implying that you don't trust the resolved scales to provide the right gustiness. Running at 3km resolution without a boundary layer above the lowest model level strikes me as extreme (not that I haven't done it myself). I would really like to see this topic directly discussed by the authors. This is a "grey zone" that seems to get less attention than the problem of transitioning from parameterized "convection" to explicit "convection" and one that I find very confusing.

We agree that the lack of boundary layer formulations in many CRMs could be a problem and has been underexplored. We anticipate that some of the CRMs participating in RCEMIP will employ boundary layer closures while others will not; it will be informative to see whether those

without boundary layer closures consistently display different behavior from those that do. If so, targeted sensitivity experiments of boundary layer formulations could be a focus of phase two of RCEMIP. We note that our inclusion of a specification for a minimum wind speed in the surface flux calculation was focused on making sure that everyone uses the *same* minimum wind speed. We now mention that models should use their default boundary layer closure, if one is employed, in our specification of the surface boundary condition (Section 3.2.1), and mention boundary layer closures as a possible focus in phase two RCEMIP simulations (Section 6).

Along the same lines, it would be nice if the text more explicitly encouraged much higher resolution simulations than 1km in the 100km domain, closer to LES resolution for the planetary boundary layer.

We agree that it would be great if modeling groups were able to complete additional simulations with higher resolution. The simulations described in the paper are only the *required* simulations – we hope that participants will be eager to go beyond them, and now explicitly state that we encourage this (see Section 3.3, first paragraph). We would like everyone to complete simulations at the 1 km resolution in the 100 km domain so that they can be compared without worrying about the impact of varying resolution. We now also explicitly encourage LES simulations in Section 3.3.5.

I would strongly encourage you to reconsider and ask groups to run with a standard microphysical mechanism in addition to their model's microphysics. Otherwise, there is a good chance that the diversity of simulations will be dominated by the diversity of microphysical assumptions. (For example, we know that in RCEs assumptions about ice fall-speeds will exert a strong control on the cirrus climate.) I realize that this would be work for the authors, who have to be confident that this standard microphysical mechanism worked reasonably well in a couple of the models that they are familiar with. The proposal here is to leave this issue to a second round, but I think this is postponing the inevitable. (For example, you suggest that models with more elaborate microphysics, with explicit activation of CCNs perhaps, be adjusted to create simulations that produce the same droplet number concentrations as recommended for models that specify these droplet numbers, but what if droplet numbers change significantly with warming in these models?)

We agree that large differences could result from differences in microphysics, and that it might be correct that the diversity of simulations will be dominated by the diversity of microphysical assumptions. However, we think that it is useful to first determine the full range of RCE simulations and *then* proceed to test the microphysics sensitivity by imposing a simple microphysics scheme on all models in the second phase of RCEMIP. In the past, groups have found large sensitivities to microphysics, but that might also reflect that the behavior of microphysical parameterizations are easiest to change (i.e., it is easy to modify a fall speed, but harder to change an underlying assumption in a boundary layer representation, for instance). In addition, it might not make sense to specify the microphysics without specifying the treatment of cloud optical properties (radiation), representation of partial cloudiness, etc…, and this would be too much to accomplish with our first set of simulations. Importantly, our goal with the first phase of RCEMIP is to keep the required simulations to a minimum and as close as possible to a models "standard" configuration so as to encourage maximum possible participation and limit

the possibility of new physics introducing bugs that are not characteristic of a specific model. Perhaps it is "postponing the inevitable" but we believe that it is worth first providing a framework and taking stock of where things stand. We think that determining, for example, how many of the models have a decrease in high cloud fraction with warming with the "standard" configuration is valuable (if hard to disentangle), because presumably all the different schemes used are individually reasonable and justified choices and we don't want to immediately bias the results in the direction of one scheme over another.

Regarding the specification of droplet number concentrations we suggested, we indicated that models should use the standard from their Aquaplanet configuration, and only provided numbers so that those without an Aquaplanet configuration would have some guidance as to what to choose.

The utility of single column models, in my view, is dependent on their "effective stability": the single column equilibrium, whether time-independent or dependent, with no resolved flow is always a solution of a fully doubly periodic box or global RCE model that uses the same column physics (in fact, this is a good test that a global model has been successfully homogenized). But it is invariably spatially unstable. To the extent that the time average of the RCE state in the 100km box bears little resemblance to the single column equilibrium, the single column results are going to be hard to relate to the rest of the project. I would try to encourage any contributions of single column RCE simulations to also provide full doubly-periodic 100km box simulations for comparison, or at least global RCE simulations if these are easier to perform.

These are important issues to consider when interpreting the results of the simulations across different model types, but we do not believe precludes the inclusion of single column models in RCEMIP. In theory, the time average of a doubly periodic box or global simulation should be the same as that of single column models; the exception is if the box or global simulation aggregates (we assume this is the spatial instability referred to above). We acknowledge that, in general, it is difficult to compare results from models that aggregate and models that do not, but we anticipate that many of the 100 km simulations at CRM resolution will not aggregate, which then should be able to be compared to the single column simulations (disagreements would result from deficiencies in either or both models, which such an intercomparison might reveal). We anticipate that most single column models will be paired with a corresponding global RCE simulation (as this is the initialization procedure for GCMs), as was suggested. And, at minimum, single column models will be able to be compared to other single column models. Overall, we simply chose to design the experiments in a way that could support take-up of RCE simulations by the single column community, should they be interested in using our framework. To clarify the SCM set-up and its utility, we have added a section about single column models in Section 3.3.4.

For the same reason, I don't particularly care for initializing global models with single column equilibria. Among other issues, what if these equilibria are time-dependent? (This is not hypothetical – it may be the rule rather than the exception with cloud- radiative interactions present). Do you initialize all the grid point with the same phase?

Part of the motivation for initializing global models with single column equilibria is to have a consistently configured "control" simulation that does not have convective aggregation to compare the global simulations to. All model grid points would be initialized with the same sounding from the single column model, but then small random noise will be added to break the symmetry. The alternative is to have global models initialize from an externally specified sounding, but we do not think this is better than using potentially time-dependent equilibria, plus, it would then be unclear how to create a control simulation without aggregation to compare to. Preliminary tests with CAM5 indicate that the RCE state achieved is not sensitive to how the model is initialized, leading us to believe that the potential issues of using single column equilibria to initialize global models will not be of first order importance.

I would not specify number of grid points to be used. I would just specify the physical size of the domain and then encourage simulations with 1km resolution in the small domain and 3km in the bowling alley domain, while also encouraging any other resolutions that the groups are interested in exploring.

We would like all models to use the same domain size and resolution so that we have a consistent configuration that can be compared across models. Given that we know that at least some aspects of RCE are sensitive to domain size and resolution (i.e., Muller and Held 2012), we do not want to conflate that sensitivity with model-model differences. We certainly do encourage additional simulations at other resolutions, with the same physical size of the domain. But we agree that it is probably better to specify the physical size rather than the number of grid points, so we have reworded the domain specification for CRMs (Section 3.2.1) to an approximate physical domain size and grid spacing (see lines 32-32, page 9 and lines 1-2, page 11).

The recommended upper boundary seems quite low to me, making it awkward to look at the vertically propagating gravity waves in the stratosphere excited by the convection. This is a nice opportunity to look at the robustness of the simulation of those waves, especially in the bowling alley case. On a related point, I would be a little worried that the bowling alley configuration will in some cases allow the generation of QBO-like oscillations, as in the two-dimensional limit. This could cause some large unexpected differences between simulations. It is worth mentioning at least.

We have changed the recommended CRM vertical grid to have a minimum of 74 levels (up to 33 km) (See line 26, page 10). The 4096 x 64 bowling alley configuration used in Wing and Cronin (2016) did exhibit QBO-like oscillations, as noted in that paper, although the effect on the troposphere was minimal. The 2048 x 128 simulation examined in that paper (which is the domain geometry to be used in RCEMIP) did not experience such strong oscillations.

You seem to be initializing stratospheric water with extremely small, but non-zero, mixing ratios. But won't stratospheric water take forever to equilibrate in RCE simulations, where the only thing that can mix vapor up is breaking gravity waves? This vapor has some effect on sensitivity of radiative fluxes to warming. Kuang and Bretherton, JAS, 2004 have circumvented this problem by specifying a small vertical motion (a Brewer-Dobson upwelling extending through the troposphere and stratosphere), which allows the tropopause cold-trap to operate. As in nature this also makes it easier to study the tropical tropopause layer, where convection and upwelling

can both be important. In any case, this issue of equilibrating stratospheric water should be discussed. (How do you get water vapor into the stratosphere in a single column model? Do you intend that these models include sub-grid closures for mixing by convectively generated waves?)

We examined the evolution of stratosphere water vapor in the preliminary CAM5 simulations and SAM simulations at several different model levels. In the CAM5 simulations, stratospheric water vapor is indeed not equilibriated by the end of year 3, but we believe this has little impact on the RCE state in the troposphere; the CAM5 simulations are in equilibrium in terms of precipitation, outgoing longwave radiation, and omega. In the cloud-permitting SAM simulations, the water vapor in the lower stratosphere seems to be approximately equilibrated within 100 days. Regarding the motivation for initializing the stratospheric water with very small but non-zero mixing ratios, we followed the procedure of Reed and Jablonowski (2011). We do not think that there is an obvious way to deal with the stratospheric water vapor correctly, and so just acknowledge in the paper that the lack of equilibration will be an issue and will be monitored and assessed in the evaluation of the simulations (see lines 15-18 on page 9).

Minor comments:

p-3, l-2. Held, Zhou, Wyman, 2007 also compared the cloud feedbacks in RCE against those in the tropics of a global model with the same column physics.

We now reference this paper (see line 8, page 3) and apologize for the omission.

p.-5, l-20. Sorry, but false precision is a pet peeve of mine. You specify the mean surface pressure to 6 significant figures, but the mass of the atmosphere is not conserved, it increase with warming due to increase in vapor. Some global models correctly incorporate the mass source/sink of vapor, and these models will just ignore the value that you specify, they need to specify the dry mass.

This has been removed in the revised manuscript.

As another example, in the list of constants you are careful to specify the value of the latent heat of vaporization, but you don't specify its temperature dependence – or, equivalently, the heat capacity of the condensate. Are you expecting the latter to be set to zero? You specify the (mean) radius of the Earth with precision and then suggest that models just use smaller radii, a good idea since the radius in the non-rotating case is totally irrelevant until it approaches the depth of the model domain. Meanwhile differences in assumptions about surface smoothness or ice fall-velocities or many other things that are literally orders of magnitude more important are passed over. The table of geophysical constants is pointless in my opinion; just ask modelers to use their standard Earth values.

Our motivation for including the table of geophysical constants was to follow the convention of other MIPs (i.e., the Aquaplanet Experiment) in providing a table of these values. We are, in fact, asking modelers to use the standard Earth values for their Aquaplanet configuration. We specified the value of the latent heat of vaporization at 0 C; models should follow their usual formulation for its temperature dependence.

I would not ask the modeling group to provide your preferred measures of aggregation. Other measures might turn out to be more interesting. These should be relatively easy to generate in a uniform way after the fact from the data archive. You should keep the post-processing requested from the groups to a minimum. (Some groups may not be interested in the dynamics of aggregation.)

We agree that the post-processing requests should be kept to a minimum and the only diagnostics that must be computed online are cloud fraction and the moist static energy budget terms. We have removed the mention of the autocorrelation length, but retain a shortened description of the two aggregation metrics, since we show plots of one of these quantities in the paper. We have rephrased the text to indicate that code to compute those metrics will be available on the website.

As for the energy budget, I am not sure why you want this in advective rather than flux form (I may be missing something.). A data request could include the fluxes of energy at the boundaries of the grid cells – at least in CRMs and grid-point GCMs. These are more likely to be precisely defined by the underlying numerical scheme than the advective form.

Previous studies have also calculated the moist static energy budget in models using the advective form (e.g., Anderson and Kuang 2012, J. Climate). To rewrite it in flux form, one needs to assume something about the continuity equation, which takes different forms in different models depending on which version of the equations of motion they use. In the end, the estimation of the advection term is going to be inherently uncertain and we just ask groups to do the best they can.

I can't see anything in Fig. 6

We are not sure what is meant by this comment; when we download the PDF of the paper from the GMD website, Figure 6 is visible. It shows the daily average column water vapor at the beginning and end of the small domain SAM simulation, which is horizontally homogeneous because the simulations are unaggregated and the convection is quasi-randomly distributed, and thus "averages out".

Response to Reviewer II (Levi Silvers) (Author Comments in Blue)
An intercomparison of models in RCE is an excellent idea. This paper provides a good framework to help organize the project and build on the current momentum that the topic of RCE has. The paper is well written. The current level of organization for this project is impressive and will hopefully lead to many participating models. The availability of diagnostic codes is also a major benefit.

We thank Dr. Silvers for his positive comments and support for RCEMIP.

I think the third theme on robustness is one of the most critical. It gets a bit lost in the discussion about aggregation and sensitivity to warming. When the literature on RCE is surveyed, the apparent diversity of results is often the primary first impression. For example, not all RCE models aggregate. The WRF model in RCE mode (for a particular configuration) does not aggregate while SAM does (see last chapter of dissertation by Wenyu Zhou). The DAM model does not spontaneously aggregate (Jeevanjee and Romps, 2013). The broad range of results displayed in Held, Zhao, and Wyman, 2007 illustrates the difficulty of establishing a 'ground-truth' or baseline for RCE experiments. Some RCE models generate a 'QBO-like' oscillation, and some have an issue with dominant and persistent upper-level clouds. Another example of large variations in RCE results is shown in Silvers et al. 2016 where the ICON model exhibits large differences in the climate sensitivity for differing domain sizes despite similar subsidence fractions. There are consistent and robust results from RCE models, and I think much can be learned from them, establishing the bounds within which RCE is consistent among models and without which RCE varies across configurations would be one of the most significant/useful scientific outcomes of RCEMIP. The authors are clearly aware of this general point (lines 16-18 of page 2), but it could be clearer in the text, and emphasizing the importance of this component could make the project more appealing to some potential participants.

It will be very useful to the community to determine the response of clouds to warming, but this will have to be interpreted through the lens of RCE and the applicability to observations will be dependent on our ability to establish a consistent picture of the RCE results. This is why I think the current first theme, clouds and climate sensitivity, is secondary to the robustness theme.

We absolutely agree about the importance of assessing the robustness of the RCE state, and have tried to emphasize this point a bit more in the revised manuscript. We have revised the abstract to more strongly state the importance of the robustness theme (Lines

7-9, Page 1), added that the difference in set up between past studies makes it difficult to determine which aspects of the simulations are robust (Lines 23-25, Page 2), note that an intercomparison can establish which features of the RCE state are consistent across models and which vary (Lines 30-31, Page 2), state that using a wider range of models is important for evaluating the generality of previous work on RCE (Line 3, Page 3), and reworded some of the description of the robustness theme in Section 2 (Lines 29-35, Page 4).

It is not essential, but I think it would be useful to be more precise about a second set of non-required (Tier 2) experiments. This could be written in such a way that modeling centers wishing to participate with minimal effort are not thus discouraged from participating,
but that more ambitious modeling centers or individuals could clearly push
farther into the project in a coordinated way. My suggestions for further experiments would be:
1. Rotating RCE 2. GCMs in RCE mode with convective parameterization turned off 3. RCE with cloud RCE off (COOKIE type experiments) 4. Kessler physics across the hierarchy of models
This doesn't necessarily need to be incorporated in the paper: Presenting the hierarchy of model types as having 3 tiers misses a critical tier of a GCM with simplified microphysics. If we are discussing tiers and hierarchies why not state five model types for RCE: LES-RCE (sub km) CRM-RCE (1-5km), DoublyPeriodic_CourseRes-RCE(simple physics), DoublyPeriodic_CourseRes-RCE(full physics), and a global GCM-RCE(full physics).

We hope that the revised manuscript (see especially Section 6) better conveys our goal with regards to the simulation design and our overall vision for the project: to keep the initial set of simulations as simple as possible in order to maximize participation, establish a baseline, and inform subsequent experimentation. We envision that RCEMIP will eventually extend far beyond the simulations laid out in this paper. However, we did not want to be too specific about a second set of simulations, as we are not sure yet what the next most important questions will be! They necessarily depend on the results from the initial simulations and it is possible that different subsets of participants will be interested in pursuing different additional lines of inquiry. Nevertheless, we have tried to provide a bit more detail and clearer organization of the themes of further experiments in Section 6, following your suggestion.

Figures:
Overall the figures are clear, well formatted, and useful. However, the current set of figures isolates the different models rather than capitalizing on the intercomparison.
It would illustrative of the motivation for RCEMIP to have a figure illustrating the same quantity across the hierarchy of model configurations. For example, it would be nice to see something like a plot of vertical mean cloud fraction from SAM, NICAM, and CAM; a panel plot showing OLR (or water vapor path) from these three model configurations; or a panel plot showing the subsidence fraction as a function of time for the three model types.

Our objective with the figures that were included was to simply show an example of what RCEMIP simulations might look like for different model types, so that participants who have never done RCE simulations below have a guideline for qualitatively determining they have set up their model correctly. We have updated the figures to show panel plots of subsidence fraction and cloud fraction for all three model types together, as suggested (see Figures 11 and 12), and plotted OLR in Figures 3 and 4 (with the same color bars as in Figures 8 and 9).

Figure 9: It isn't a big deal, but why weren't the RCEMIP protocol parameters used for this figure?
This is a sample figure from a simulation performed before the development of the RCEMIP protocol, included here to show an example of what RCE simulated by a GCRM might look like.

Figures 12-14: I am assuming that these figures are showing data from the same 3 CAM simulations. Am I right? If so, this is not explicitly mentioned in the captions, and should be.
Yes, they are showing data from the same three CAM simulations, we have corrected this in the caption.

Specific Comments:
Page 2:
Lines 16-18: This seems to say that the reason it is still unclear if the observed atmosphere aggregates is because details of aggregation in models are dependent on model formulation. But in my opinion the lack of clarity on aggregation in observations and its relevance to climate change is mostly due to the difficulty of connecting an RCE model to cases with strong dynamics and horizontal gradients of forcing.

Yes, of course, the real world is much more complex than RCE, but we wanted to emphasize that even *if* we knew how to connect RCE to more complex cases, it would still be hard to do because some aspects of the simulations are sensitive to how the models are formulated. That is why we used the phrase "in part".

Line 27: I agree that it will be useful to extend the range of models used to simulate RCE, but single-column models have already been used to study, or in coordination with, RCE. So single-column models don't extend the range that already exists, nor will the proposed experiments here. What RCEMIP will do in this regard is to help fill-out the ensemble space of a baseline set of model configurations. This is important to evaluate the generality ('robustness') of the previous work on RCE, and will help to identify or map out what types of RCE models are still needed in the hierarchy to better our understanding.

Yes, it is correct that single-column models have already been used to study RCE; in fact, as we mention in the Introduction (Lines 15-16, Page 1), much of the history of modeling

RCE has been in a single column! Part of the motivation for designing RCEMIP such that single column models can participate is to connect back to that history, and here, to some extent, they will be able to be *compared* to other model types in a consistent set up. We have removed the sentence that was referenced and instead list all the model types to which the RCE framework is accessible (lines 34-35, Page 2) and stating that using a wide range of models is important for evaluating the generality of previous work on RCE (lines 2-3, Page 3).

Line 30: Citing Silvers et al. here is questionable, as we did not make the comparison suggested, rather out intention was to motivate the comparison. All of our domain sizes had the same grid-spacing. Page 2, line 2 would be more appropriate. Also page 4, line 32 (on comparing the spread of climate sensitivity) would work, but is not necessary.

We have changed the citation of Silvers et al 2016 to Line 5, Page 2 as suggested.

Concerning the themes outlined, themes 2 and 3 seem quite broad and as a result a bit vague. Perhaps broadness is the aim?

We have slightly revised the themes to be
1. What is the response of clouds to warming and the climate sensitivity in RCE?
2. What is the dependence of convective aggregation and tropical circulation regimes on temperature in RCE?
3. What is the robustness of the RCE state, including the above results, across the spectrum of models?

The themes are meant to be broad guidance as to what RCEMIP will address; more specific lines of inquiry are detailed in the paragraphs following the identification of the three themes (pages 3-5).

Lines 2-4: This sentence seems to be overly emphasizing what RCE would tell us about clouds in the observed atmospheric system. The connection between the climate sensitivity of RCE simulations and the fundamental characteristics of modeled clouds is almost clear, but what are the fundamental characteristics of modeled clouds, perhaps you could say the climate sensitivity is one of those characteristics but not all encompassing? My interpretation is that the RCE convection is a key baseline that will help us understand the role of modeled convection in determining the climate sensitivity. It would be useful to ask for the ice-fall speed, and the fraction of convective precipitation from models to be saved. These would be helpful to categorize or interpret seemingly different results (as discussed in Held, Zhao, and Wyman, 2007).

Requesting the output of fraction of convective precipitation is a good idea– we now request the convective precipitation rate from GCMs in Table 5 as "pr_conv". Ice-fall speed would also be useful, but may be too "non-standard" of a variable to request (and is often simply a parameter).

Technical Corrections:
There should be a "to" on line 9 between "guide" and "those who"
This has been corrected, thank you.

Line 14: Perhaps delete this comment? 'mimicking' sounds like you are playing tricks here, or like you are not really representing the state described. I think something like, 'representing' would be more accurate. You are representing the described state. Shortcomings in the representation come from RCE, or the model itself, not from the boundary conditions.
We have changed this to "representing", as suggested.

Line 24: 'conditions' should be singular.
This has been corrected, thank you.

Page 10:
Line 28: "from" should be "for", "with", or "in".
This has been corrected, thank you.

Line 33: There is an extra "to" in the last line.
This has been corrected, thank you.

Should more be said about necessary adjustments when altering the radius of earth? Maybe just citing more of Kevin Reed's work would be an easy way to give people a clue about the details without adding too much technical material to the paper.
Reed and Medeiros (2016) is already cited as an example of reduced Earth radius being used in RCE studies.

Page 13:
Line 2: Should "aggregation" be added here to indicate what was compared? As written, it is a bit unclear and the sentence should be reworded a bit for those who have not read Cronin and Wing.
We now specify that Tompkins and Semie (2017) and Cronin and Wing (2017) examined $I_{org}$ in the context of simulations of self-aggregation.

Page 14:
Line 17: 'Figure' should be plural.
This has been corrected, thank you.

Page 15:
Should we really be calling a GCM with 14 km horizontal grid-spacing a cloud resolving simulation (GCRM)?

It is a fair point that a grid spacing of 14 km does not properly resolve clouds, but the relevant point is that the model is run without a convective parameterization; this type of

model has been referred to as a GCRM. However, we have rephrased the sentence in question to read: "…an example result from a global simulation of RCE with explicit convection, using NICAM in a global, spherical configuration with real Earth radius and a 14-km horizontal grid spacing."

Please refer to Section 3.4 "GCRMs" for our definition of GCRMs and their experimental protocol.
Overall, I am very supportive of the initiative and recommend publication of this protocol paper in the non-discussion GMD journal. The discussion paper is largely focused on aggregation questions, which are a pressing scientific concern that this intercomparison will be critical to addressing, but there is also a lot of value in the small domain simulations (what does the cloud fraction look like there? e.g., Fig. 12). So, some additional discussion of the value of the MIP independent of aggregation questions would be well justified. I look forward to seeing the science enabled by RCEMIP.

We thank the reviewer for their positive and constructive comments.

We have tried to emphasize the importance of assessing the robustness of the RCE state more in the revised manuscript. We have revised the abstract to more strongly state the importance of the robustness theme (Lines 7-9, Page 1), added that the difference in set up between past studies makes it difficult to determine which aspects of the simulations are robust (Lines 23-25, Page 2), note that an intercomparison can establish which features of the RCE state are consistent across models and which vary (Lines 30-31, Page 2), state that using a wider range of models is important for evaluating the generality of previous work on RCE (Line 3, Page 3), and reworded some of the description of the robustness theme in Section 2 (Lines 29-35, Page 4).

Major points:

(i) I found it odd that the presentation of the preliminary results of the intercomparison were separated by model. Why not actually compare, for example, the OLR of the simulations in the same figure with the same color bar (vs. Fig. 3, 9, 10)? Even if the authors decide to leave the separate structure (SAM then NICAM then CAM5), the figures should allow for an "apples-to-apples" comparison (precipitation rate intervals and colors differ between Fig. 9 and 10, for instance). Last, Fig. 12 should either have analogous results from other models or not be included.

Our objective with the figures that were included was to simply show an example of what RCEMIP simulations might look like for different model types, so that participants who have never done RCE simulations below have a guideline for qualitatively determining they have set up their model correctly. We have updated the figures to show panel plots of subsidence fraction and cloud fraction for all three model types together (see Figures 11 and 12), ensured the same color bars for precipitation rate intervals, and plotted OLR in Figures 3 and 4 (with the same color bars as in Figures 8 and 9).

(ii) The analytic initial condition for the small domain simulation needs to be motivated. It's only necessary if multiple equilibria are simulated. Otherwise, it's an additional barrier to participation.

We agree that the initial condition shouldn't matter for the equilibrium state, but participants need to choose something as the initial profile for the small domain/single column simulations and so we wanted to provide guidance as to what they should choose, and determined an analytic profile would be easier to implement than asking participants to interpolate a provided sounding, and that fitting an observed sounding was most justifiable. This methodology also follows Reed et al 2015. We have revised the description of the initialization procedure to reflect this (see Lines 7-11, Page 8).

(iii) Description of domains: p.10: Is it correct that only a rectangular channel geometry is part of the MIP for CRMs? This is a non-obvious choice, so it would be good to state clearly that there is no square domain simulation requested. Do you think it's best to specify approximate horizontal resolution and a fixed number of grid points in the two directions? I would prefer the opposite: like Lx = 6000km, Ly = 400km.

A set small square domain simulations ARE requested for CRMs. They will serve as the initialization simulations for the larger channel domain and as a non-aggregating control; this is detailed in section 3.2.3 and 3.3.1.  To clarify this, we have we reworded the list of required simulations (Section 3.1) to be two sets of simulations at 3 different SSTs; "RCE_small" and "RCE_large" (see Lines 17-29, Page 5 and Lines 1-4, Page 6).

We have also changed the domain specification to specify an approximate domain length/width and grid spacing as suggested, to allow groups the flexibility to choose the precise dimensions that work with their model setup (see Lines 31-32, Page 9 and Lines 1-2, Page 11).

Also, GCRMs are still non-rotating? I wasn't sure how to interpret L20 says "run on a sphere". This is like the CAM simulations shown: non-rotating but Earth's geometry, right?

Yes, all simulations are to be non-rotating, the statement in question was meant to distinguish using Earth geometry from a doubly periodic planar geometry. An example GCRM simulation with NICAM is shown which is non-rotating but with Earth geometry. The phrase has been changed to say "run on a non-rotating sphere" to clarify.

(iv) Aggregation metrics: I was unsure about the need for some of these to be pre- computed by the participants. The Organization index proposed is determined by the OLR distribution, so that's something that isn't necessary to pre-compute. Likewise for the subsidence fraction, unless there is something about the temporal frequency of the output that I missed. In contrast, I definitely understand that it's valuable if the modeler/modelling center provides moist static energy budgets.

We agree that the post-processing requests should be kept to a minimum but think what we ask for is reasonable. We have removed mention of the autocorrelation length, but retain a shortened description of the two aggregation metrics, since we show plots of one of these quantities in the

paper. We have rephrased the text to indicate that code to compute those metrics will be available on the website so that it is easy for everyone to compute them.

 (v) Climate sensitivity and connection to more comprehensively configured GCMs: Perturbing SST allows for a quantification of the "Cess-sensitivity", but the 5 K interval for the SST perturbation is larger than uniform warming simulations in comprehensive MIPs. So, please provide additional motivation for this choice. I believe it may also be "big" from the aggregation perspective (Wing et al 2014 had big changes from unaggregated to near-peak aggregation for a comparable magnitude perturbation).

The motivation for the 5K intervals is to cover a wide temperature range with a limited number of simulations, which is now stated in Line 1 on Page 6 of the revised manuscript. We also mention that optional simulations at intermediate SSTs or warmer or cooler SSTs could be performed by modeling groups if desired.

The other aspect of climate sensitivity that this kind of intercomparison (with both GCMs and CRMs) could address is tropospheric cloud adjustments to changing CO2. It would be good to see how GCMs compare to CRMs in this aspect of their sensitivity (see, e.g., Wyant et al. 2012 JAMES doi:10.1029/2011MS000092). It's also an important complement to narrowly comparing the Cess sensitivity between RCE and Earth boundary conditions. The RCE configuration may have a different radiative forcing because of the differences in the control simulation cloud distribution or differences in cloud adjustments. For example, compared to GCMs, RCE may have a big Cess-sensitivity (in the large warming sense), but also a smaller radiative forcing.

We now mention an abrupt $4xCO_2$ experiment as a possible simulation in the second phase of RCEMIP (see Lines 1-2 of Page 22, in Section 6).

Minor comments:

* author affiliations out of order 4 and 5

This has been corrected, thank you.

* p. 1 L4 "role of self-agg..." on what?

This has been re-phrased to say "role of self-aggregation in climate sensitivity"

* single column models -> single-column models (though maybe conventional compound-adjective practices aren't used for SCM)

I believe the convention is to use "single column models".

* p.1 L15 climate sensitivity estimates: expect a Manabe and Weatheral 1967 reference here

This has been added, thank you.

\* p. 2 L9 a couple of earlier RCE precip extremes papers from Muller, Romps

Citations of Muller et al 2011 and Romps 2011 have been added.

\* p. 2 "formulaic sensitivity" I found this confusing

This has been changed to say "structural sensitivity".

\* p. 6 L 29 "the lapse rate" -> " the virtual temperature lapse rate"

This has been changed as suggested.

\* p. 8 after going through the description of the continuous analytic formula for the initial condition, a discrete near-surface perturbation is used (lowest 5 layers); the perturbation seems irrelevant to GCMs, unless I'm missing something.

The purpose of the near-surface perturbation is to break the symmetry and trigger convection quickly.

\* p. 10 L 12 missing punctuation after GCM

This has been corrected, thank you.

\* p.11 L21 stray )

This has been corrected, thank you.

\* p.12 Do you want to specify the variable names for the horizontal coordinates of the CRM output? If participants are forced to convert variables names to the CMOR format, it would be good also do something consistent for the coordinates (I'm not suggesting labelling them latitude and longitude, to be clear).

It might be unavoidable that the coordinate names are different for CRM and GCM output, but it should be fairly self-explanatory.

\* p.18 Fig 8 caption: what are whiskers? standard deviation? Some of the simpler aggregation metrics should be evaluated in the non-SAM simulations (see major issue 1)

The error bars indicate the bounds of the 5-95% confidence interval, and we now plot subsidence fraction for SAM, NICAM, and CAM simulations (see Figure 12).
The authors suggest an intercomparison project for various types of models run in radiative-convective equilibrium (RCE). While previous studies have differed in details of their setup the aim of this paper is to provide a common baseline. First, the setup of the intercomparison is detailed, then some sample results are provided.
Overall, I think that this is a great initiative and the suggested setup useful. There are a few suggestions to consider:

We thank the reviewer for their positive and constructive comments.

1. As e.g. detailed in Wing et al. (2017), the resulting equilibrium state and the clustering may look very different in the different CRMs. It will thus be very difficult to compare the different models and to identify the root for the differences (radiation scheme, microphysics parametrizations, . . .). An even simpler setup for the models could therefore be useful to identify, which schemes are responsible for the differences. As suggested by the authors and brought forward by Jeevanjee et al. (2017) a simplified microphysics scheme could be one option. A further option could be to simplify the longwave radiative cooling, as e.g. described in Muller and Bony (2015). Such a simplified simulation should be run as number 0 at one SST to assess science objective 3. If this simulation already showed large differences between the individual CRMs running simulations 1-3 should be reconsidered.

We agree that large differences could result from differences in physical parameterizations. However, we think that it is useful to first determine the full range of RCE simulations and *then* proceed to test the parameterization sensitivity by imposing a simple microphysics scheme on all models in the second phase of RCEMIP. In the past, groups have found large sensitivities to microphysics, but that might also reflect that the behavior of microphysical parameterizations are easiest to change (i.e., it is easy to modify a fall speed, but harder to change an underlying assumption in a boundary layer representation, for instance). As implied by the reviewers comment, it might not make sense to specify the microphysics without specifying the treatment of cloud optical properties (radiation), representation of partial cloudiness, etc…, and this would be too much to accomplish with our first set of simulations. Importantly, our goal with the first phase of RCEMIP is to keep the required simulations to a minimum and as close as possible to a models "standard" configuration so as to encourage maximum possible participation and limit the possibility of new physics introducing bugs that are not characteristics of a specific model. We believe that it is worth first providing a framework and taking stock of where things stand. We think that determining, for example, how many of the models have a decrease in high cloud fraction with warming, with the "standard" configuration is valuable (if hard to disentangle), because presumably all the different schemes used are individually reasonable and justified choices and we don't want to immediately bias the results in the direction of one scheme over another.

2. A number that is hardly discussed in the aggregation literature is the heating/cooling rate at which the equilibrium is reached. Which longwave cooling rate is balanced by convection? How much latent heat is released? This will again be reflected in the surface precipitation rate. How strongly does this number differ between simulations and what is its value in observations? The output from multiple models would give an indication of how much this value varies, and how GCMs with parametrized convection compare with CRMs/observations. A value of ~100 W/m2 for precipitation is found, but how much variation around this value is there between the participating models? On page 4, last paragraph it is mentioned that radiative flux divergence is nearly a universal function of temperature, which in turn is a function of temperature only. Before investigating the response of RCE to warming it is important to first focus on the robustness of these fundamental quantities across models.

These are good ideas of things to quantify when investigating the robustness of the RCE state, thank you for the suggestions; we now mention some of them when describing the ``robustness'' theme in Section 2.

3. A strong focus of the project will lie on the coupling between convection and circulation. At the same time it is stressed that the proposed framework will be suitable for SCMs, that are unable to resolve circulations by design. Moreover, aggregation can not occur. Please specify more clearly which aspects from the SCMs will be analyzed and how they can contribute to a deeper understanding of aggregation in RCE.

The SCMs will be used to investigate some of the questions about robustness of the RCE state, such as the quantities mentioned in your above comment, and will serve as a control for the responses in a model *without* aggregation. This will serve as an important comparison to a GCM with the same physics that could support aggregation (as mentioned in Section 3.2.3). In theory, the time average of a doubly periodic box or global simulation should be the same as that of single column models; the exception is if the box or global simulation aggregates. To clarify the SCM set-up and its utility, we have added a section about single column models in Section 3.3.4.

Specific comments:
page 2, line 14: correct "explict"
This has been corrected, thank you.

page 3, line 4: remove "in" before "between"
This has been corrected, thank you.

page 3, line 9: add "to" before "those"
This has been corrected, thank you.

page 10, last paragraph: a grid spacing of 60 or even 220 km very likely not produce reasonable convection, which makes the acquired RCE state questionnable.
This has been removed, thank you.

page 10, line 33: remove "to"
This has been corrected, thank you.

page 13, line 21: replace "estimate" with "estimates"
This has been corrected, thank you.

page 20, line 4: please list the unanswered questions laid out by Wing et al., (2017) for those readers who are not very familiar with the paper.

We now summarize the unanswered questions laid out by Wing et al. (2017) (see Lines 9-10, Page 21).

Figures 9 and 10: please give the point in time

This has now been included in the captions.
Apologies for the late post.... About a half dozen GFDL scientists, along with two GFDL post-docs, met to discuss the RCEMIP proposal and GFDL's potential participation using the FV3 dynamical core. This note outlines some of the thoughts and responses that came up.

There is a common interest here in developing a doubly-periodic, cloud-resolving RCE configuration of FV3 which uses our current ('AM4') comprehensive physics package and which would be suitable for RCEMIP. Motivations for this are diverse, however. Probably the most common motivation is to use RCE as an idealized testing ground, for use in (say) comparing microphysics schemes, or benchmarking low-resolution parameterized-convection simulations against high-resolution explicit convection simulations on the same domain. Note that these activities have little to do with RCEMIP as currently presented, though there is some mention (but little emphasis) in the manuscript on planar GCM configurations.

A second motivation for our development of FV3 RCE would be to assess how idiosyncratic our simulated, unaggregated RCE state is relative to other models. This falls nicely in line with the 'robustness' objective of RCEMIP (science objective #3), though (as other reviewers have pointed out) this objective seems to get de-emphasized in the paper. The authors may want to consider increasing their emphasis on it.

Convective self-aggregation seems to be a major focus of RCEMIP, and while there is some interest in aggregation here at GFDL, overall it is probably only a secondary concern. Thus, while interest in a suite of kilometer (or even sub-kilometer) small- domain RCE simulations is strong, interest in aggregated simulations (and especially the computation of secondary, aggregation-focused diagnostics) seems to be weaker.

As advocated for by Isaac and other reviewers, there is also interest here in using simplified (Kessler) microphysics in our RCE setup. Such a scheme already exists in development branches of our code.

We thank Dr. Jeevanjee for his comments and are pleased that GFDL scientists are interested in RCEMIP, and we sincerely hope that GFDL will contribute one or several configurations of the FV3 model to RCEMIP.

We hope that the revised manuscript (see especially Section 6) better conveys our goal with regards to the simulation design: to keep it as simple as possible in order to maximize participation, establish a baseline, and inform subsequent experimentation. We envision that RCEMIP will eventually extend far beyond the simulations laid out in this paper, as we recognize that the baseline we propose will not be a definitive representation of the RCE climate

for many of the reasons raised by the reviewers. While it is clear that certain physical parameterizations (or lack thereof) may lead to biases that require further investigation, we see the simulations proposed as a way to bring the community together to get us to that next point. We believe one of the strengths of RCEMIP is the sheer number of scientific questions that can be investigated within this framework, and while we have described the motivations as we see them, there is certainly room for other interests, and we in fact encourage groups to use this framework and the intercomparison as a tool to explore their individual interests. We do mention in both the introduction and conclusions that RCEMIP will serve as a useful framework for model development and evaluation, and in Sections 3.1 and 3.3 encourage modeling groups to simulate both RCE on the sphere and plane. We now also explicitly encourage LES simulations in Section 3.3.5. We have also clarified what the proposed simulations are in Section 3.1, describing them as a set of small domain simulations and a set of large domain simulations.

We absolutely agree about the importance of assessing the robustness of the RCE state, and have tried to emphasize this point a bit more in the revised manuscript. We have revised the abstract to more strongly state the importance of the robustness theme (Lines 7-9, Page 1), added that the difference in set up between past studies makes it difficult to determine which aspects of the simulations are robust (Lines 23-25, Page 2), note that an intercomparison can establish which features of the RCE state are consistent across models and which vary (Lines 30-31, Page 2), state that using a wider range of models is important for evaluating the generality of previous work on RCE (Line 3, Page 3), and reworded some of the description of the robustness theme in Section 2 (Lines 29-35, Page 4).

We also recognize that post-processing requests should be kept to a minimum but think what we have asked for is reasonable. Only the cloud fraction and moist static energy budget terms need to be computed online. We have removed the mention of the autocorrelation length, but retain a shortened description of the two aggregation metrics, since we show plots of one of these quantities in the paper. We have rephrased the text to indicate that code to compute those metrics will be available on the website to make it easy for everyone to compute them.

Regarding the simplified microphysics scheme, we are very supportive of applying such a scheme in the second phase of RCEMIP. We agree that large differences could result from differences in microphysics, and that it might be correct that the diversity of simulations will be dominated by the diversity of microphysical assumptions. However, we think that it is useful to first determine the full range of RCE simulations and *then* proceed to test the microphysics sensitivity by imposing a simple microphysics scheme on all models in the second phase of RCEMIP. In the past, groups have found large sensitivities to microphysics, but that might also reflect that the behavior of microphysical parameterizations are easiest to change (i.e., it is easy to modify a fall speed, but harder to change an underlying assumption in a boundary layer representation, for instance). In addition, it might not make sense to specify the microphysics without specifying the treatment of cloud optical properties (radiation), representation of partial cloudiness, etc…, and this would be too much to accomplish with our first set of simulations. Importantly, our goal with the first phase of RCEMIP is to keep the required simulations to a minimum and as close as possible to a models "standard" configuration so as to encourage maximum possible participation and limit the possibility of new physics introducing bugs that are not characteristics of a specific model. We believe that it is worth first providing a

framework and taking stock of where things stand. We think that determining, for example, how many of the models have a decrease in high cloud fraction with warming, with the "standard" configuration is valuable (if hard to disentangle), because presumably all the different schemes used are individually reasonable and justified choices and we don't want to immediately bias the results in the direction of one scheme over another.

---

## Author Response (AR2)

Response to Anonymous Referee #3 (Author Comments in Blue)

Minor revisions:

*The second sentence (p.1 L14) would be clear with commas between atmosphere and convection and between atmosphere and making.

This has been corrected as suggested.

*p. 3 L7 maybe "approximately moist adiabatic thermal structure"

This has been corrected as suggested, to "an approximate moist adiabatic thermal structure"

*p. 4 L3 cloud fraction changes do not determine the feedback parameter even in RCE. No change in cloud fraction but a change in cloud water path would be a non-zero feedback. Please correct.

Yes, you are absolutely correct, and we did not mean to imply that the change in cloud fraction would literally be used to compute the net feedback parameter, but see now how that sentence could have been interpreted that way. We meant that the changes in clouds contribute to the climate sensitivity (as implied in the previous sentence). We have changed the text to say "The net feedback parameter of the RCE state may be computed, which is reminiscent of…."

*p. 5 L 10 "truly determine" the word choice of "truly": these are all model sensitivities being discussed, so there isn't an observable truth.

That is a fair point, so we have changed the phrase to "better determine".

*Figures do not all have the axes dimensions consistently labeled (e.g., 5 vs 4)

Figures 3, 4, 5, and 6 now all have consistent axis labels. We have also corrected a typo in the title of Figures 3 and 4.

*It's arguably implicit in the discussion of extensions to RCEMIP, but a natural extension would be to isolate/simplify radiative effects to get a baseline of the mean state of convection permitting vs. parameterized convection simulations (goal 3) in the absence of things like cloud radiative effects that are one source of differences. This sort of goes back to the reaction to the initial submission's focus on aggregation. If you want to study aggregation, removing some of the interactivity of the radiative cooling side is undesirable, but if you want to see how parameterized convective tendencies fair relative to CRMs, it would be interesting.

Yes, we agree that a set of simplified/non-interactive radiation simulations would be valuable, and we do mention in Section 6 (p. 20 L11-12) that a suite of simulations with cloud radiative effects turned off could be performed. In response to your comment, we have added more detail to that suggestion to make the objective more explicit. The sentence now reads "For example, a suite of simulations with cloud radiative effects turned off could be performed, which would be

useful for comparing the mean state of simulations with explicit to that of those with parameterized convection, in the absence of self-aggregation."

Response to Anonymous Referee #4 (Author Comments in Blue)

Review of revised version of
"Radiative-Convective Equilibrium Model Intercomparison Project"
by Allison A. Wing, Kevin A. Reed, Masaki Satoh, Bjorn Stevens, Sandrine Bony, and Tomoki Ohno

The authors have put considerable effort to into answering to the review. They have replied to all of the points to my satisfaction. I thus recommend to accept the article for publication.

There are a few technical points that should be corrected:

Thanks for finding these errors!

Page 8, line 15: there is something odd with the specific humidity at the surface q_0, it is higher in the simulations at 300K than for the simulations at 305 K. I guess it should be increasing with temperature?

This was a typo, q0 for the simulation at 305 K should be 24 g/kg, which we have corrected.

Page 13, lines 16 and 25: there is a spurious parenthesis after coordinates.

This has been corrected.

Page 14, line 17: replace "off" by "of"

This has been corrected.

Page 18, line 10: include "be" after "can"

This has been corrected.

[revised manuscript text omitted]